# *Explore Theory of Mind*: PROGRAM-GUIDED ADVERSARIAL DATA GENERATION FOR THEORY OF MIND REASONING

**Melanie Sclar**[1,2,*]   **Jane Yu**[1]   **Maryam Fazel-Zarandi**[1]
**Yulia Tsvetkov**[2]   **Yonatan Bisk**[3,*]   **Yejin Choi**[4]   **Asli Celikyilmaz**[1]

[1]FAIR at Meta  [2]University of Washington  [3]Carnegie Mellon University  [4]Stanford University
msclar@cs.washington.edu

## ABSTRACT

Do large language models (LLMs) have theory of mind? A plethora of papers and benchmarks have been introduced to evaluate if current models have been able to develop this key ability of social intelligence. However, all rely on limited datasets with simple patterns that can potentially lead to problematic blind spots in evaluation and an overestimation of model capabilities. We introduce EXPLORETOM, the first framework to allow large-scale generation of diverse and challenging theory of mind data for robust training and evaluation. Our approach leverages an A* search over a custom domain-specific language to produce complex story structures and novel, diverse, yet plausible scenarios to stress test the limits of LLMs. Our evaluation reveals that state-of-the-art LLMs, such as Llama-3.1-70B and GPT-4o, show accuracies as low as 0% and 9% on EXPLORETOM-generated data, highlighting the need for more robust theory of mind evaluation. As our generations are a conceptual superset of prior work, fine-tuning on our data yields a 27-point accuracy improvement on the classic ToMi benchmark (Le et al., 2019). EXPLORETOM also enables uncovering underlying skills and factors missing for models to show theory of mind, such as unreliable state tracking or data imbalances, which may contribute to models' poor performance on benchmarks.[1]

## 1 INTRODUCTION

Reasoning about other people's intentions, goals, thoughts, and beliefs is a foundation of social intelligence. Known as *Theory of Mind* (ToM) (Premack & Woodruff, 1978), this capability is crucial for effective human interaction. There has been a plethora of recent research that develops theory of mind benchmarks and test LLM capabilities, usually inspired in standard tests for research in children such as the Sally-Anne test (Wimmer & Perner, 1983). However, these tests are not well-suited for extensively evaluating models, as they focus on specific scenarios and lack the variability and complexity required to remain challenging after online pre-training. As a result, many existing computational benchmarks may not be effective in robustly evaluating models' theory of mind abilities.

We introduce **EXPLORETOM**, an A*-powered algorithm for generating reliable, diverse, and challenging theory of mind data that can be effectively employed for testing or fine-tuning LLMs. Our approach leverages a domain-specific language to generate synthetic story structures and their character's mental states. We then use LLMs to create plausible stories based on these plots, allowing for precise control over the narrative and tracking each character's mental state with high confidence. We employ A* search (Hart et al., 1968) to efficiently navigate the vast space of possible narratives and pinpoint those that are most likely to fool state-of-the-art LLMs. This in turn allows to create a robust, rich dataset that effectively tests the limits of current models (Fig. 1). By generating story structures separately from lexical realizations, we can distinguish the model's core understanding of the social reasoning from vocabulary cues that might give away stylistic hints.

Our contributions are three-fold: we algorithmically address blind spots in theory of mind evaluation, we provide a recipe to create complex training data that helps imbue models with better theory of mind reasoning skills, and we provide insights into why theory of mind skills are still elusive for LLMs.

---

*Work done at Meta. Results presented here reflect latest results collected up to Dec. 12 2024.

[1]Code and data can be found at https://github.com/facebookresearch/exploretom.

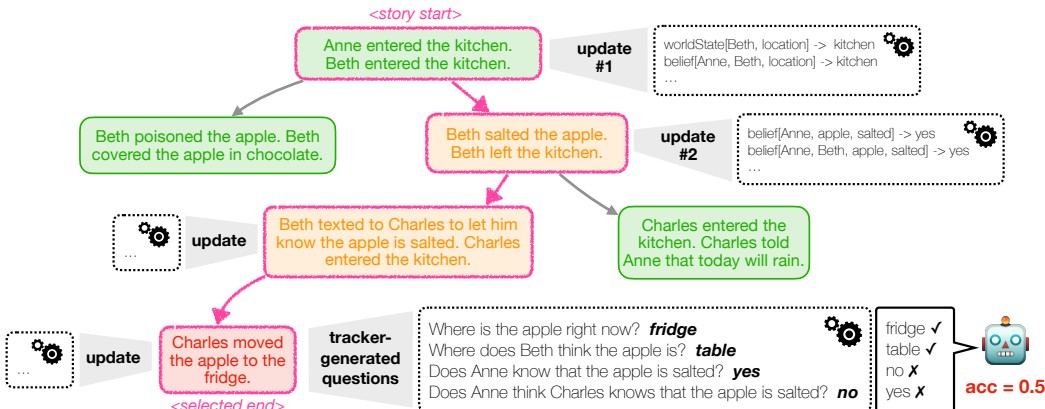

Figure 1: EXPLORETOM finds challenging stories for a given language model by searching through the space of stories supported by its domain-specific language for mental state tracking (⚙), sampling $k$ supported actions at a time (shown as a node, $k = 2$ in the example). Difficulty evaluation (simplified in the figure as easy, medium, hard) of each partial story is done through automatically generated questions with reliable ground-truth answers thanks to our tracking procedure.

First, our work helps to address the conflicting results from a large number of prior research on evaluating theory of mind (e.g. Sap et al., 2022; Shapira et al., 2023a; Kim et al., 2023; Zhou et al., 2023; Gandhi et al., 2024; Strachan et al., 2024), including reports that due to oversimplified datasets, prior theory of mind estimates may be overly optimistic (Ullman, 2023). Our algorithmic approach also helps address the issue of developing benchmarks that may be close to saturation at the time of release, given the increasingly harder task of anticipating LLM failures (e.g., Kim et al., 2024). EXPLORETOM's adversarial nature allows for generating stories to stress test any LLM, diminishing the risk of data leakage onto training data, and thus being more robust than manually-crafted benchmarks. Our experiments show that **EXPLORETOM-generated data is extremely challenging, with GPT-4o and Llama-3.1 70B accuracies as low as 9% and 0% respectively.** EXPLORETOM supports significantly more scenarios than previously possible, with the unique addition of knowledge gain asymmetry during an interaction, among other improvements.

Second, our method allows creating complex and diverse theory of mind data that can be leveraged for model fine-tuning. Given theory of mind's implicit nature, it is challenging to find data that explicitly articulates the required reasoning, and existing benchmarks are not suitable to use as training data: they are often limited in scale (Xu et al., 2024), portray specific scripted scenarios (Wu et al., 2023; Le et al., 2019), and are prone to leakage risks that would make them fully unsuitable for future use. Fine-tuning with this data has been shown to overfit to specific story structures instead of learning the underlying reasoning required (Sclar et al., 2023), leading to works focused on creating inference-time algorithms to improve the model's capabilities through prompting or more sophisticated strategies (Sclar et al., 2023; Zhou et al., 2023; Wilf et al., 2023; Jung et al., 2024). While inference-time methods have proven useful for improving performance in theory of mind benchmarks, the benefits of these methods cannot be readily transferred to downstream applications that may also require customized inference-time algorithms for their specific use cases. Fine-tuning Llama-3.1 8B Instruct on EXPLORETOM's data achieves a substantial **+27 accuracy point improvement on the classic ToMi benchmark** (Le et al., 2019), **showing good generalization to even more complex EXPLORETOM stories than those seen during training, while still retaining general reasoning capabilities**.

Finally, EXPLORETOM enables providing new insights into why basic theory of mind reasoning is still challenging for LLMs. We show that LLMs struggle with basic state tracking, a fundamental skill underlying theory of mind reasoning: tracking mental states necessarily requires being able to track states. Our experiments also reveal that in order to improve on theory of mind during fine-tuning, it is crucial to use data that requires theory of mind as opposed to simply requiring state tracking. However, found data (either in-the-wild, or randomly generated) is unlikely to have this necessary property, which may be a key contributor to lagging model performance. Overall, EXPLORETOM offers a valuable tool for advancing the theory of mind research, enabling the development of more effective LLMs that can better handle complex social interactions.

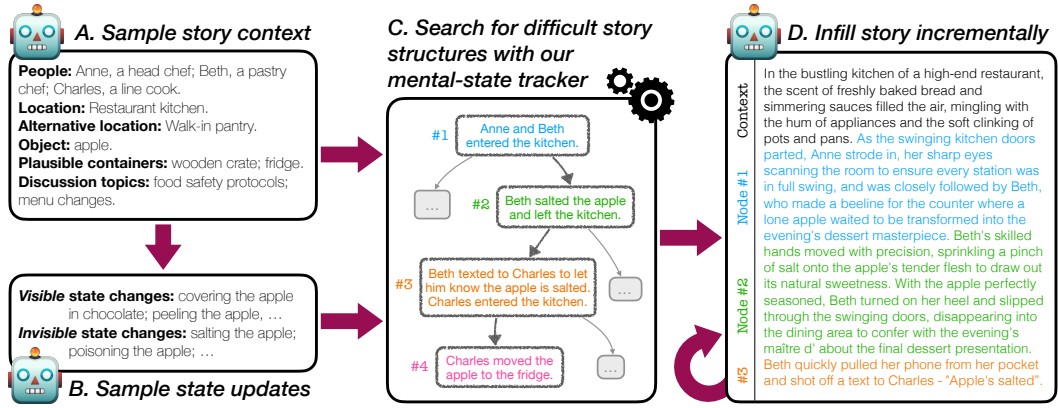

Figure 2: Overview of EXPLORETOM's story generation procedure. We first sample a plausible story context using an LLM (shown in A and B). Topics discussed, location changes of objects and people, and object state updates, may all be required to track in order to pass our theory of mind tests. We then search for difficult story structures (i.e., the raw story points) by sampling and analyzing different orders in which these actions may be performed using A* search (shown in C, and Fig. 1). This ensures that the resulting stories will all be challenging tests for models, and may be used for further improvement. Finally, these story structures (nodes #1-4) are iteratively infilled, one story action at a time, using a language model, yielding a natural-sounding story. Infilled stories are used as training data; benchmarking is done with story structures since they have the highest reliability.

## 2  ADVERSARIALLY CONSTRUCTED STORIES WITH EXPLORETOM

Building on the standard approach in theory of mind of assessing mental state understanding through question answering (Wimmer & Perner, 1983; Kinderman et al., 1998; Baron-Cohen et al., 1999), EXPLORETOM creates stories where different characters may have different beliefs about the current world state and about other people's beliefs, paired with questions to probe model understanding (see Fig. 1's highlighted story, along with associated questions probing understanding that e.g. "Anne does not know that Charles knows that the apple has been salted").

EXPLORETOM's story generation process is divided into three main steps: plausible story context sampling (Section 2.1), adversarial story structure generation (Section 2.2), and optionally story infilling (Section 2.3) – an example is outlined in Figure 2. We automatically generate questions to probe understanding of said stories as part of the adversarial story structure generation process (Section 2.2.2); this process finds challenging story structures, i.e., story structures that would yield low accuracy with our generated questions. Because questions are generated automatically and directly from the tracked mental and world states, ground truth answers have a high degree of reliability: we do not use language models at all in the question-answer generation procedure.

### 2.1  PLAUSIBLE STORY CONTEXT SAMPLING

We use an LLM zero-shot to generate a consistent and plausible story context, comprising essential elements such as character names, roles, locations, relevant objects, object containers, and discussion topics (see Fig. 2A for a full example). This single-step process ensures a coherent and believable setup for our theory of mind stories. Previous approaches (such as ToMi (Le et al., 2019)) sample objects (e.g., an apple) and object containers independently (e.g. a bottle), often resulting in commonsense violations. Unlike these approaches, our method generates a coherent context by sampling these elements jointly in a single LLM call: autoregressive LLMs will naturally suggest contextually plausible elements based on the ones they already generated, and especially so when explicitly requesting it in the prompt. Additionally, we sample possible object state updates (Figure 2B), which are then refined through using an LLM as a judge to filter out implausible and low quality generations. The role of these state updates will be discussed in further detail in Section 2.2.1. The exact prompts used for sampling story contexts are shown in App. C.

## 2.2 Adversarially Generating Challenging yet Plausible Story Scripts

### 2.2.1 Theory of Mind-Specific Language Definition

EXPLORETOM's theory of mind-specific language consists of a diverse set of actions $\mathcal{A}$, each transforming the world state and the beliefs of the people involved (the story *state* $s \in \mathcal{S}$). A *story* is thus defined as a sequence of actions $(a_1, \ldots, a_n)$, where each action $a_i \in \mathcal{A}$ is a function $a_i : \mathcal{S} \to \mathcal{S}$. Each action also has preconditions to be able to apply it, i.e., restrictions to its domain. For example, a precondition for "Charles entering the kitchen" is to not be in it already. Applying an action also automatically updates our world state tracking and belief tracking: for example, "Charles is now in the kitchen"; "Anne knows that Charles is in the kitchen since they were also in the kitchen"; "Charles knows that Anne is in the kitchen since he can see her"; and so forth. All these updates and conditions are specifically programmed and tested; see App. A.1 for the full programs.

EXPLORETOM enables the generation of diverse stories by significantly expanding the range of supported actions. These actions include physical changes to the world state such as entering and leaving a room (denoted $a_{\text{enter}}$, $a_{\text{leave}}$), moving an object to a container (or in general, updating its state; denoted $a_{\text{moveObjContainer}}$, $a_{\text{updateObjState}}$ respectively), relocating an object to a different room ($a_{\text{moveObjRoom}}$). Additionally, EXPLORETOM supports various forms of communication, including: private conversations between two characters, or public broadcasts to all characters in a room; casual discussions about a topic (denoted *chit-chat*), or notifications about changes in the world state (denoted *info*); these actions are referred to as $a_{\text{info-private}}$, $a_{\text{info-public}}$, $a_{\text{chitChat-private}}$, and $a_{\text{chitChat-public}}$. These actions can occur at any point in the story, allowing for a rich and dynamic narrative (see formal definition in App. A.1) and expanding prior work (Wu et al., 2023).

Each new action requires carefully writing the implied belief and world state updates, which precludes scaling the number of actions supported. However, we alleviate this by noting that from a theory of mind perspective, many actions are equivalent. For example, "peeling an apple" or "covering an apple in chocolate" have the same implications with respect to belief updates (a *visible property* of the apple is being updated, and the witnesses would be the same). Similarly, poisoning an apple has the same implications as moving an apple from a drawer to a fridge (an *invisible property* is updated, witnesses would be the same, and non-witnesses would not assume there has been an update). The instantiations of these equivalent state updates from a belief perspective are done with an LLM during the story context sampling (see Figure 2.B).

**Asymmetric belief updates**   In prior work, all belief updates were *symmetric*: if A and B witnessed an action, then A knows that B witnessed the action and vice versa. Our framework introduces the ability to model asymmetric scenarios. Specifically, we enable the addition of secret witnesses to an action such as someone observing through a security camera, or removal of witnesses without others' knowledge, as in the case of someone becoming distracted by their phone. This added nuance allows for more realistic and complex social scenarios. Asymmetries $a_{\text{peek}}$ and $a_{\text{distracted}}$ are modifier functions, e.g., as a modifier to "Beth salted the apple" ($a_{\text{updateObjState}}(\cdot)$) there may be a secret person peeking ($a_{\text{peek}}(a_{\text{updateObjState}}(\cdot))$): "While this was happening, Diane witnessed it in secret."

### 2.2.2 Generating Questions and Assessing Resulting Story Difficulty

We assess a model's understanding of a generated story $s=(a_1, \ldots, a_n)$ by probing it with automatically generated question-answer pairs. EXPLORETOM-generated answers are more reliable than purely-LLM generated ones, since they are directly produced from the states' trajectory with our tracker. Questions may be testing first-order beliefs, second-order beliefs, or regular state tracking: *First-order* refers to asking about someone's mental state (e.g., "Does Anne know the apple is salted?"); *Second-order* refers to one extra level of recursion in mental state tracking (e.g., "Does Anne think that Charles know the apple is salted?"); *State tracking* may probe about the current state (*ground truth*) or prior ones (*memory*).

We expand the complexity of memory questions with respect to prior work by asking about any intermediate state (e.g. "Where was the object before X happened?") instead of solely about the initial one ("Where was the object at the beginning?"). Our generated questions are simple to evaluate: they are either binary (yes/no), or are answered by stating an object, container, or room. Specific question formulations differ based on the property, e.g., location ("Where does Charles think that

Anne will search for the apple?") or knowledge ("Does Charles know that the apple is salted?"). See App. A.2 for the full list of supported questions.

A question is considered *interesting* if the answer would change depending on the person being asked about. For example "Does Anne think that Charles knows that the apple is salted?" is interesting because the answer would differ if asked about someone else, such as "Does Beth think that Charles knows the apple is salted?". EXPLORETOM's tracker very easily allows for automatically detecting interestingness.

### 2.2.3 A* SEARCH

Given a context $\mathcal{C}$ and a set of actions $\mathcal{A}$, our main goal is to find challenging story structures. To increase EXPLORETOM's usage flexibility, we support the option of searching for stories $s$ that fulfill desired user conditions **isDesired**$(s) \in \{0, 1\}$, such as the number of people involved, or the number of actions belonging to a subset $\mathcal{A}' \subseteq \mathcal{A}$ of *important actions*.

We search over the space of plausible story structures of up to $m$ actions. We define this space as a directed graph, where each node is a sequence of valid actions $s = (a_1, \ldots, a_i)$, and there is an edge between $s$ and $s'$ if and only if $s$ is prefix of $s'$, and $s'$ contains $k$ more actions than $s$. $k \geq 1$ is the *grouping factor* for actions, defining the granularity with which we will sample and evaluate nodes. For simplicity, Figure 1 depicts only the new $k = 2$ actions that each node introduces.

To find challenging stories that simultaneously fulfill the user constraints we use A* search (Hart et al., 1968). By definition, A* selects the path that minimizes $f(s) = g(s) + h(s)$, where $g(s)$ is the cost of the path from the start to node $s$, and $h(s)$ is a heuristic that estimates the cost of the cheapest path from $s$ to a *goal node* (one of the nodes where it would be acceptable to finish the search). In our context, goal nodes are those such that **isDesired**$(s') = 1$. We choose A* as our search algorithm precisely because it enables to search this space prioritizing desired user conditions through $h(s)$, as we will detail below.

A story is said to be challenging for a model if it incorrectly answers our generated questions, i.e., it shows low accuracy. Thus, we define $g(s)$ as our target model's accuracy among *all* questions for $s$. We define the heuristic function $h(s)$ as a proxy estimation of the likelihood of generating a full story $s + s'$ that fulfills user constraints **isDesired**$(s) = 1$, where $s'$ is the continuation of story $s$:

$$h(s) = \alpha \Big(1 - \frac{1}{P} \sum_{i=1}^{P} \mathbb{1}(\textbf{isDesired}(s + s'_i) = 1)\Big)$$

Here, all $s'_i$ are randomly sampled continuations of $s$ and $0 \leq \alpha \leq 1$ is a scaling factor. A* requires to evaluate all neighbors of a node $s$. Since this would be infeasible given the vast space to explore, and that each $f(\cdot)$ evaluation requires several LLM calls (one per question), we restrict the evaluation to a pre-defined constant number of neighbors, prioritized by the closeness of this node to fulfilling the conditions described by **isDesired**$(\cdot)$. This pre-defined constant may depend on $f(s)$ to prioritize more promising partial stories (i.e., with lower $f(s)$ values).

### 2.3 STORY INFILLING

*Story infilling* is the process of transforming a full story structure $s = (a_1, a_2, \ldots, a_n)$ with a story context $\mathcal{C}$ into a natural-sounding narration (see Fig. 2D). We infill stories iteratively with an LLM by transforming each action $a$ into a more natural sounding one, according to some stylistic desiderata $d$, and conditioned on the previously infilled context $z$ (denoted infill$(a, z, d)$). Supported stylistic desiderata $d$ are length requests (e.g., "use up to two sentences") or style requests (e.g., "make this into a conversation"); we optionally also include sampled character goals $g$ and an initial narration context $c$ based on the story $s$, also generated with an LLM (e.g., Anne's goal may be to oversee that all dishes are rapidly delivered to customers; see initial context example in Fig. 2). Concretely, the full story infilling $SI$ is as follows:

$$SI(i) = \text{infill}(a_i, SI(i-1), d_i, g) \text{ where } SI(0) = c$$

Infilling is done iteratively to ensure that the order of the actions stays the same, since this is important for keeping the mental state tracking valid. To further increase reliability, we use an LLM as a judge after each infilling step to confirm that each mental state tracked after executing the story step $a_i$ still holds even after infilling. This discards infillings that introduced ambiguity or hallucinations.

Table 1: Accuracy results of EXPLORETOM's story structures on 18 action sets $\mathcal{A}$, each aggregating 90 total stories from 9 different settings (number of people, actions, and rooms). Each set is either based on actions supported by well-known theory of mind tests or includes our novel expansions, and is analyzed excluding or including asymmetry (✗, ✓). Each setting requires at least one action in the story to be from one of the $\boxed{\text{squared}}$ actions to encourage non-overlapping story structure characteristics between action sets shown. Data was generated using each model as its own evaluator (i.e., as $g(\cdot)$), and results shown include all first-order questions—the most basic theory of mind level, not requiring recursion. Lowest accuracy for each model is bolded.

| EXPLORETOM action set $\{a_{\text{enter}}, a_{\text{leave}}, \ldots$ | Llama-3.1 70B Inst. | | GPT-4o | | Mixtral 8x7B Inst. | |
|---|---|---|---|---|---|---|
| include asymmetry modifiers? ($a_{\text{peek}}, a_{\text{distracted}}$) | ✗ | ✓ | ✗ | ✓ | ✗ | ✓ |
| $\ldots, \boxed{a_{\text{moveObjContainer}}}\}$ | .18 | **.00** | .40 | .25 | .37 | .33 |
| $\ldots, \boxed{a_{\text{updateObjState}}}\}$ | .27 | .24 | .25 | .24 | .03 | .01 |
| $\ldots, \boxed{a_{\text{moveObjContainer}}}, \boxed{a_{\text{updateObjState}}}\}$ | .26 | .03 | .35 | .31 | .24 | .09 |
| $\ldots, a_{\text{moveObjContainer}}, \boxed{a_{\text{moveObjRoom}}}\}$ | .11 | .10 | **.09** | **.16** | **.00** | **.00** |
| $\ldots, a_{\text{moveObjContainer}}, \boxed{a_{\text{info-*}}}\}$ | **.06** | .07 | .29 | .25 | .35 | .36 |
| $\ldots, a_{\text{moveObjContainer}}, a_{\text{moveObjRoom}}, \boxed{a_{\text{info-*}}}\}$ | .11 | .07 | .24 | .24 | .03 | .04 |
| $\ldots, a_{\text{moveObjContainer}}, a_{\text{moveObjRoom}}, a_{\text{chitChat-*}}, \boxed{a_{\text{info-*}}}\}$ | .72 | .69 | .73 | .68 | .53 | .47 |
| $\ldots, \boxed{a_{\text{chitChat-private}}}\}$ | .75 | .66 | .77 | .59 | .51 | .45 |
| $\ldots, \boxed{a_{\text{chitChat-public}}}\}$ | .60 | .55 | .49 | .47 | .37 | .34 |

Figure 3: Accuracy on EXPLORETOM's story structures when increasing the number of actions or people involved. Accuracy is computed across all story structure settings. Difficulty of EXPLORETOM-generated stories tends to increase or stay similar when increasing the number of actions. A story with greater number of people suggests similar or lower difficulty, possibly because when fixing the number of actions there are fewer actions per person (see details in App. B.5).

## 3 EXPLORETOM AS AN EVALUATION BENCHMARK

We begin by showcasing how EXPLORETOM story structures can be used as a challenging benchmark, highlighting its unique features and advantages.

**Experimental setup** We use EXPLORETOM to generate 10 story structures for each of 9 action sets (each with and without asymmetry) and each set of user conditions. Each story generation is allowed to evaluate 50 nodes. User conditions—isDesired($\cdot$)—require exactly $p \in \{2, 3, 4\}$ people involved, with $a \in \{2, 3, 4\}$ actions belonging to the set of important actions $\mathcal{A}'$, spanning across either $r = 1$ or $r = 2$ rooms, and with $m \leq 15$ actions in total—leading to a total of 162 settings. In all experiments, $\mathcal{A}'$ are the actions that add new basic world knowledge: $\mathcal{A}' = \{a_{\text{moveObjContainer}}, a_{\text{updateObjState}}, a_{\text{moveObjRoom}}, a_{\text{chitChat-*}}\}$. We then infill every story. We use Llama-3.1-70B-Instruct (Dubey et al., 2024), GPT-4o (OpenAI (2024); queried early Dec. 2024), and Mixtral-8x7B-Instruct (Jiang et al., 2024) to generate story structures. A* is run with $\alpha = 0.1$, $P = 50$, and $k = 3$ (i.e. grouping three actions per node). See generation examples in App. D.

**EXPLORETOM finds challenging story structures for frontier models** As shown in Table 1, our EXPLORETOM consistently identifies story structures that are highly challenging for models

Table 2: Accuracy results on EXPLORETOM-generated data built to minimize accuracy for each particular model. A random sample of 1000 (story structure, question) pairs is shown. Data remains challenging even if it was built with a different model, and even including questions we did not optimize for: story structures were selected adversarially towards first-order belief questions only ($g(\cdot)$), accuracies shown include all belief questions.

| | Model used for evaluation | | |
|---|---|---|---|
| Model used in EXPLORETOM generation ($g(\cdot)$) | Llama-3.1 70B Inst. | GPT-4o | Mixtral 7x8B Inst. |
| Llama-3.1 70B Inst. | 0.57 | 0.68 | 0.32 |
| GPT-4o | 0.60 | 0.61 | 0.32 |
| Mixtral 7x8B Inst. | 0.68 | 0.74 | 0.30 |

across various action sets, with average performances in EXPLORETOM-generated datasets as low as 0.09 for GPT-4o (i.e., 9%). When increasing the number of actions, difficulty tends to increase or remain similarly challenging. Performance tends to stay the same or decrease when increasing the number of people involved, possibly because with a fixed number of state-changing actions there will be fewer actions per person which may be easier to track. See Figure 3 and App B.5.

**A\* is a better strategy than over-generation and filtering**    Over-generation and filtering has become a standard procedure for synthetic data generation (e.g. West et al., 2022; Wang et al., 2023). We measure the effectiveness of A\* by comparing the A\*-generated data to the data resulting from over-generating stories with our domain-specific language—using the same **isDesired**($\cdot$) criteria and budget as used in the A\* search—and retaining only the most difficult stories. In a set of 81 randomly-selected settings (50% of the original 162 settings, due to the experiment's high cost), we generate 50 stories with each method using Llama-3.1-70B-Instruct and a budget of 2500 accuracy evaluations each. A\* yielded a more challenging dataset (by 2 accuracy points), with shorter stories on average (1.6 fewer actions). This length difference is possibly due to the pressures A\* induces towards shorter stories through the heuristic $h(s)$. See Figure 6 for the full distribution of results.

**Story structures found adversarially for a model remain challenging for other models**    We evaluate the difficulty of a EXPLORETOM-generated dataset with each model, and find that although there is an increased difficulty towards data generated adversarially with the same model, it remains challenging for all others. Notably, the generated datasets remain challenging even when adding question types not included in the $g(\cdot)$ optimization (second-order belief questions). See Table 2.

**Humans agree with EXPLORETOM-generated story structures labels**    We conducted a human evaluation to verify the quality of the story structures' automatically-generated labels and the story infillings. For labels, we annotated 100 questions across 12 randomly-sampled story structures from all settings generated for Table 1, and found 99% agreement with our expected answers—likely due to the clear and concise nature of our stories and that the ground truth labels were generated by our domain-specific language. We measure story infilling quality by repeating the question-answering procedure with a different set of 100 questions across 12 randomly-sampled infilled story structures. In this case, the human agreed with the ground truth label 89% of the time—a small degradation likely due to the LLM-powered method introducing ambiguity.

**Infilled stories remain challenging**    Infilled stories with Llama-3.1 70B yielded an average accuracy of 0.61. Although the average accuracy increased by 0.12 through the infilling process, the samples remained challenging thanks to the highly challenging underlying stories[2]. One key factor for this accuracy difference comes from models sometimes making the mental states more explicit through the infilling process: results shown correspond to a single attempt at infilling each story (41% of the samples ended successfully in a single attempt, judged by an LLM). Although stories remain challenging, since infilling with an LLM may introduce some ambiguities or hallucinations we only use them as training data. See App. B.2 for detailed results for all action sets.

---

[2]Infilling can be also added to the A\* search; we deemed it unnecessary given that this simpler method still yields a highly challenging benchmark and it is less costly.

Table 3: Performance on major false-belief benchmarks; accuracy (in %) unless otherwise stated. Parenthesis reflect differences between out-of-the-box model and fine-tuned version using EXPLORETOM-generated data. **Bold** reflects higher overall performance.

|  | ToMi | Hi-ToM | BigToM | OpenToM (F1) | FANToM |
|---|---|---|---|---|---|
| Llama-3.1 8B Instruct | 68% | 30% | 75% | .39 | 0.3% |
| EXPLORETOM-8B | **95%** (+27) | **59%** (+29) | **81%** (+6) | **.46** (+.07) | 0.2% (-0.01) |

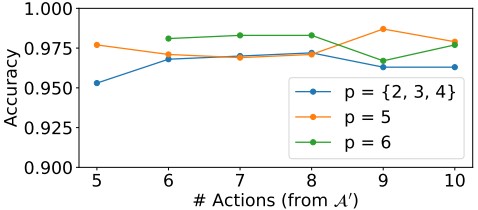

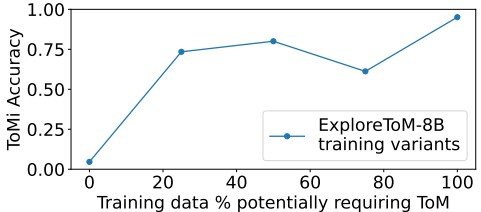

Figure 4: EXPLORETOM-8B accuracy when evaluating on EXPLORETOM-generated data with more people $p$ and/or more actions $a$ than seen during training ($p < 5, a < 5$). Performance remains high when adding several actions and/or up to two people.

Figure 5: ToMi accuracy when training with EXPLORETOM-generated data with different proportions of interesting questions (i.e., questions potentially requiring theory of mind to answer). Here, all variants are fine-tuned with 85000 story structure samples for 1 epoch.

## 4    EXPLORETOM IS EFFECTIVE AS TRAINING DATA GENERATOR

**Experimental setup**    We fine-tune Llama-3.1 8B Instruct using a dataset of 79700 (story, question, answer) triples, focusing solely on the completion tasks, and dub the resulting model EXPLORETOM-8B. The dataset comprises both raw story structures and infilled stories, incorporating story structures from each of the 9 action sets listed in Table 1 (excluding asymmetry, and with a balanced number of questions per story type), and various user constraints—the same as in Section 3. We do full fine-tuning with the following hyperparameters: a learning rate of $10^{-6}$, 100 warm-up steps, effective batch size of 40 samples, where we fine-tune solely on completions.

**Fine-tuning with EXPLORETOM generalizes well to EXPLORETOM-generated data with more people and more actions than used in training**    Since EXPLORETOM-8B is trained with EXPLORETOM-generated data involving $p = \{2, 3, 4\}$ people with $m = \{2, 3, 4\}$ actions from the set of important actions $\mathcal{A}'$, we evaluate generalization within the EXPLORETOM domain by evaluating on EXPLORETOM-generated data involving 5 people, and up to 11 actions. This data is generated with Llama-3.1, the same model as original training data. See Figure 4.

**Fine-tuning with EXPLORETOM improves or maintains performance on theory of mind benchmarks without hurting general reasoning capabilities**    We evaluate our fine-tuned EXPLORETOM-8B model on five prominent theory of mind benchmarks: ToMi (Le et al., 2019), Hi-ToM (Wu et al., 2023), BigToM (Gandhi et al., 2024), OpenToM (Xu et al., 2024), and FAN-ToM (Kim et al., 2023). Results show significant improvements in performance on ToMi and HiToM, with accuracy gains of +27 points on both benchmarks (see Table 3). The model maintains or shows small gains on the remaining three similar benchmarks, indicating that fine-tuning on EXPLORETOM data enhances or preserves performance across a range of theory of mind tasks[3].

We also evaluate out-of-domain reasoning skills using the two datasets: Multi3Woz (Hu et al., 2023), a commonly-used dataset for dialogue state tracking, and MMLU (Hendrycks et al., 2021), which tests both world knowledge and problem-solving abilities. Dialogue state tracking capabilities are preserved: both the base model and EXPLORETOM-8B achieve 96%. Broader reasoning capabilities are also generally preserved, with a small 2% performance difference (base model achieves 69%; EXPLORETOM-8B, 67%). Given the out-of-domain nature, we expect that intermixing data with samples more similar to MMLU's domains will substantially alleviate this slight regression.

---

[3]The lack of performance difference in FANToM is likely due to its length confounding factor: data points can have 1000+ tokens, yet our model is fine-tuned with a maximum length of 300 tokens.

**Data mixture affects downstream performance**     We fine-tune five models, with 0%, 25%, 50%, 75%, or 100% of the stories requiring theory of mind to answer at least one question about the story. Figure 5 shows that training with as much stories that require theory of mind is crucial for achieving high downstream performance (using ToMi as a proxy dataset), even if some of the individual questions used for training do not require theory of mind.

## 5   ON UNDERLYING SKILLS NEEDED FOR THEORY OF MIND

EXPLORETOM enables uncovering and quantifying underlying causes for models' poor theory of mind reasoning in models out-of-the-box. We specifically focus on the lack of robust state tracking skills, and the need for targeted training data in order to improve theory of mind capabilities.

**LLMs lack robust state tracking skills**     EXPLORETOM's objective is to find story structures where models fail to answer questions; some of these questions simply require state tracking, specifically the ones where every person would give the same answer (i.e., their mental state is the same in this regard; e.g., in Fig. 1, all $X \in \{\text{Anne, Beth, Charles}\}$ would answer the same to "Where does X think Anne is right now?"). By definition (see § 2.2.2), these are the *uninteresting* questions. EXPLORETOM-generated questions are approximately evenly split between interesting and uninteresting, and uninteresting ones are even more challenging on average: the accuracy of interesting and uninteresting questions is 49% and 31% respectively for Llama-3.1 70B, 58% and 37% for GPT-4o, and 45% and 26% for Mixtral. See Table 6 in App. B.3 for full breakdown for all settings.

State tracking questions are a subset of theory of mind questions, and arguably an easier case since the required logic for answering questions is simpler. Therefore, improving models' performance on state tracking may be a crucial prerequisite for achieving theory of mind reasoning in LLMs. As we have demonstrated, EXPLORETOM can be easily adapted to stress test pure state tracking, simply by retaining only the uninteresting questions.

**Training data biases against theory of mind and its implications**     Figure 5 shows that to successfully improve performance on the ToMi benchmark, EXPLORETOM fine-tuning data needs to be biased towards *interesting* questions. However, a significant portion of models' training data is likely biased against requiring the tracking of divergent mental states (e.g., news articles).

As a conceptual proof that this phenomena occurs even within our custom domain-specific language unless we explicitly bias towards theory of mind, we demonstrate that randomly-sampled story structures tend not to require theory of mind. Using EXPLORETOM's domain-specific language, we randomly generate 1000 story structures with ToMi primitives ($\{a_{\text{enter}}, a_{\text{leave}}, a_{\text{moveObjContainer}}\}$) for stories involving $\{2, 3, 4\}$ people and $\{2, 3, 4\}$ object movements. We consider a story to not require theory of mind if all first-order and second-order theory of mind questions are *un-interesting*, as defined in § 2.2.2 (i.e., all share the same mental state). This stringent criterion evaluates *all* questions simultaneously. Nevertheless, our results show that 78% or more of the randomly-sampled stories meet this condition across all settings, with up to 87% of stories fulfilling the condition for the smallest setting (2 people, 2 object movements). When considering each question individually, 91%-95% are uninteresting questions. See App. B.4 for more details.

## 6   RELATED WORK

**Theory of mind benchmarking for language models**     Theory of mind benchmarks in language models can be categorized into human-generated and model-generated datasets. While human-generated datasets (Shapira et al., 2023b; Kim et al., 2024; Chen et al., 2024) test reasoning about goals, emotions of others, and future actions, they are often limited in size and scope. Machine-generated datasets, such as foundational ToMi (Le et al., 2019) and its successor Hi-ToM (Wu et al., 2023) focus primarily on mental state tracking, but have significant limitations: ToMi only supports a restricted set of actions ($\{a_{\text{enter}}, a_{\text{leave}}, a_{\text{moveObjContainer}}\}$), while Hi-ToM adds $a_{\text{info-}*}$ but only as the last action in a story, and both datasets have extremely restricted interactions to orders. In contrast, our method, EXPLORETOM, significantly expands the scope of machine-generated datasets by supporting a larger number of actions, diverse wording, and plausible contexts. Unlike recent approaches that rely on LLMs for generation (Kim et al., 2023; Xu et al., 2024; Gandhi et al., 2024), EXPLORETOM ensures reliability and multi-interaction storytelling, making it a more comprehensive and robust benchmark for theory of mind in LLMs.

**Theory of mind beyond language modeling** Theory of mind has been explored in various areas, including human computer interaction (Wang et al., 2021), explainable AI (Akula et al., 2022), and multi-agent reinforcement learning (Rabinowitz et al., 2018; Sclar et al., 2022; Zhu et al., 2021). Recent benchmarks have evaluated theory of mind in multi-modal settings (Jin et al., 2024) and multi-agent collaboration (Bara et al., 2021; Shi et al., 2024), but these focus on goal-driven interactions. Psychologists distinguish between affective (emotions, desires) and cognitive (beliefs, knowledge) theory of mind (Shamay-Tsoory et al., 2010), with cognitive theory of mind developing later in children (Wellman, 2014). Our work targets cognitive theory of mind, which is well-suited for generating situations with a domain-specific language and provides unambiguous answers across cultures. By focusing on cognitive theory of mind, our approach complements existing research and provides a comprehensive benchmark for this crucial aspect of human reasoning in language models.

**Synthetic data generation** Synthetic data has become promising approach for acquiring high-quality data in various domains, including multihop question-answering (Lupidi et al., 2024), and language model evaluation (Wang et al., 2024). The process involves data augmentation/generation and curation, with techniques such as permutation-based augmentation (Yu et al., 2024; Li et al., 2024a) and iterative prompting (Yang et al., 2022). However, model hallucination (Guarnera et al., 2020; Van Breugel et al., 2023; Wood et al., 2021; Zhang et al., 2023) requires careful filtration and curation to ensure data quality. While prior works have used external feedback (Zelikman et al., 2022; Luo et al., 2024), our approach leverages an external LLM-as-judge to evaluate the plausibility and challenge of generated stories, both before and after infilling. Recently, AutoBencher (Li et al., 2024b) has also been proposed to automatically search for datasets that meet a salience, novelty, and difficulty desiderata, highlighting the importance of careful benchmark creation. Unlike AutoBencher, which over-generates under the assumption that text-based conditioning minimizes hallucinations, our approach lifts this assumption and actively searches the space of possible narratives. This enables to create high-quality synthetic data regardless of the likelihood of a story being generated zero-shot, and generating even more challenging stories than with over-generation.

# 7 CONCLUSIONS

Theory of mind (ToM) is essential for social intelligence, and developing agents with theory of mind is a requisite for efficient interaction and collaboration with humans. Thus, it is important to build a path forward for imbuing agents with this type of reasoning, as well as methods for robustly assessing the of models' theory of mind reasoning capabilities.

We present EXPLORETOM, an A*-powered algorithm for generating reliable, diverse and challenging theory of mind data; specifically, creating synthetic stories that require theory of mind to understand them, along with questions to probe understanding. EXPLORETOM's adversarial nature enables the stress testing of future models and making our evaluation more robust to data leakage. We show that EXPLORETOM generates challenging theory of mind evaluation sets for many frontier models, with accuracies as low as 0% for Llama-3.1 70B Instruct and 9% for GPT-4o. Moreover, we show that EXPLORETOM can be used as a method for generating training data, leading to improvements of up to 29 accuracy points in well-known theory of mind benchmarks. Synthetic data is crucial for this domain, given that data that articulates theory of mind reasoning is difficult to find in the wild: children have access to a wide range of naturalistic social settings that incentivize the development of theory of mind but there is no such parallel pressure for LLMs.

Finally, we provide insights as to why basic theory of mind is still elusive to LLMs, including poor state tracking skills and demonstrating the need for training data that purposefully requires theory of mind, which is likely not present in the wild nor in randomly-generated data.

## LIMITATIONS

EXPLORETOM offers a valuable tool for theory of mind research, and is a first step towards developing LLMs that can handle social interactions effectively. Although its data encompasses diverse and challenging settings—more than previously available—, and is grounded in established psychological tests, EXPLORETOM necessarily simplifies the complexity of real-world states and narratives by constraining it to the supported types of actions and interactions. Our framework requires manual coding of new actions, wich can be time-consuming process but comes with the benefit of a significant reliability improvement. Furthermore, our stories are not necessarily goal-oriented narratives, highlighting an important avenue for future work: creating datasets where actions stem directly from character goals to further enhance diversity and plausibility.

## ACKNOWLEDGMENTS

We thank Ansong Ni for the valuable discussions. We gratefully acknowledge the Meta AIM fellowship support for Melanie Sclar. This material is based upon work partly funded by the Defense Advanced Research Projects Agency (DARPA) under Contract No. FA8650-23-C-7316 and ONR N00014-24-1-2207 Gift from Amazon. Any opinions, findings, conclusions, or recommendations expressed in this material are those of the authors and do not necessarily state or reflect those of the United States Government or any agency thereof.

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

# A  APPENDIX

## A.1  ACTIONS' FORMAL DEFINITION (CONT. FROM 2.2.1)

All actions are functions that transform a state into another state, updating the world state and the beliefs of everyone involved up to two levels of recursion. All actions have preconditions, e.g. to enter a room you need to not be in it already.

A state $\in \mathcal{S}$ is comprised of a world state $ws$ (the things currently true physically about the world described), the first-order beliefs $b_1$, and the second-order beliefs $b_2$. First-order beliefs describe what each person believes to be the current world state, e.g. Anne believes that the apple is salted. Second-order beliefs describe what each person estimates that each other person believes to be the current world state, e.g. Anne believes that Beth thinks that the apple is salted.

Let's describe the definition of leaving in a room through an example: "Beth left the kitchen.", and build the definition of the action function $a_{\text{leave, Beth, kitchen}} : \mathcal{S} \to \mathcal{S}$. As described above, the state is comprised of a world state, first-order beliefs, and second-order beliefs, i.e.,

$$a_{\text{leave, Beth, kitchen}}(ws, b_1, b_2) := (ws', b_1', b_2')$$

Let's first describe the world state update $ws'$. The world state remains the same for every entity (object, container, person, etc.), except for the person leaving the room—Beth. Thus,

$$ws(q, \text{room}) = ws'(q, \text{room}) \ \forall q \neq \text{Beth} \quad \text{and} \quad ws'(q, \text{room}) = \neg\text{kitchen}$$

Let's then describe the first-order belief updates $b_1'$. Here, we assume that everyone in the same room as Beth (the kitchen) will know that Beth has left. We denote this group of people as witnesses(kitchen):

$$\text{witnesses(kitchen)} := \{p | ws(p, \text{room}) = \text{kitchen}\}$$

Everyone not in the kitchen will assume that Beth is still there unless communicated otherwise, since they have no reason to believe she has left. Thus,

$$b_1(p, \text{Beth}, \text{room}) = b_1'(p, \text{Beth}, \text{room}) = \text{kitchen} \ \forall p \notin \text{witnesses(kitchen)}$$

$$b_1(p, \text{Beth}, \text{room}) = \neg\text{kitchen} \ \forall p \in \text{witnesses(kitchen)}$$

We now describe the second-order belief updates $b_2'$. Here, we assume that everyone in the kitchen (including Beth) assumes that everyone else in the kitchen knows Beth left (and only them). If someone was not in the kitchen, they will assume nothing has happened. Formally,

$$b_2(p, q, \text{Beth}, \text{room}) = b_2'(p, q, \text{Beth}, \text{room}) = \text{kitchen} \ \forall p \notin \text{witnesses(kitchen)}, \ \forall q$$

$$b_2(p, q, \text{Beth}, \text{room}) = \neg\text{kitchen} \ \forall p \in \text{witnesses(kitchen)} \ \forall q \in \text{witnesses(kitchen)}$$

$$b_2(p, q, \text{Beth}, \text{room}) = \text{kitchen} \ \forall p \in \text{witnesses(kitchen)} \ \forall q \notin \text{witnesses(kitchen)}$$

Finally, the function can only be applied if Beth is in the kitchen, i.e. it has the precondition $ws(\text{Beth}, \text{room}) = \text{kitchen}$.

All other functions definitions can be found verbatim in the code to be released.

## A.2 ALL SUPPORTED QUESTIONS (CONT. FROM SECTION 2.2.2)

Table 4: List of all supported EXPLORETOM questions per property discussed and level of theory of mind, transcribed verbatim.

| Property asked about | ToM Order | Question (requesting *Short Answer.* in prompt) | Expected Answers |
|---|---|---|---|
| room location | – | In which room was the <object> at the beginning? | room name |
| room location | – | In which room is the <object> now? | room name |
| room location | – | In which room was the <object> before <action>? | room name |
| room location | 1st | In which room will <person> search for the <object>? | room name |
| room location | 2nd | In which room does <person1> think that <person2> will search for the <object>? | room name |
| container location | – | In which container was the <object> at the beginning? | container name |
| container location | – | In which container is the <object> now? | container name |
| container location | – | In which container was the <object> before <action>? | container name |
| container location | 1st | In which container will <person> search for the <object>? | container name |
| container location | 2nd | In which container does <person1> think that <person2> will search for the <object>? | container name |
| abstract topic knowledge | 1st | Does <person1> know about <topicDiscussed>? | *yes* or *no* |
| abstract topic knowledge | 2nd | What does <person1> think about <person2>'s belief on <topicDiscussed>? (knows about it / does not know about it) | *knows about it* or *does not know about it* |
| knowledge about state update | 1st | Does <person> believe that the <object> <newState>? Answer yes or no. | *yes* or *no* |
| knowledge about state update | 2nd | Does <person1> believe that <person2> believes that the <object> <newState>? Answer yes or no. | *yes* or *no* |

# B ADDITIONAL EXPERIMENTS

## B.1 A*-GENERATED STORIES ARE MORE CHALLENGING THAN OVERGENERATING AND FILTERING (CONT. FROM § 3)

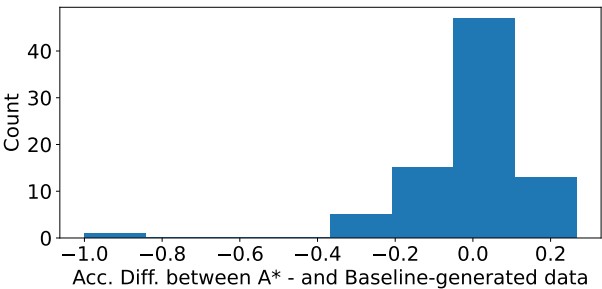

Figure 6: Histogram depicting accuracy differences between A*-generated datasets for Llama-3.1-70B-Instruct and a dataset created by over-generating and filtering with the same budget (i.e., baseline). Results show that A* is better at finding story structures that make a challenging benchmark by showing low accuracy (negative values mean A* is better at finding challenging story structures).

## B.2 INFILLED STORY STRUCTURES REMAIN CHALLENGING (CONT. FROM § 3)

Table 5: Changes in accuracy when infilling EXPLORETOM-generated story structures to output natural-sounding stories. We only include comparison between the 40% of stories where the LLM as a judge (Llama-3.1-70B Instruct) determined that all infilled actions were high quality.

| Action Set: $\{a_{enter}, a_{leave}, \ldots$ | Include asymmetry | Acc. Story Structure | Acc. Infilled | Acc. Diff. |
|---|---|---|---|---|
| $\ldots, a_{moveObjContainer}\}$, (denoted $\mathcal{A_1}$) | ✗ | 0.20 | 0.43 | 0.22 |
| | ✓ | 0.02 | 0.36 | 0.35 |
| $\ldots, a_{updateObjState}\}$, (denoted $\mathcal{A_2}$) | ✗ | 0.40 | 0.44 | 0.03 |
| | ✓ | 0.49 | 0.48 | -0.01 |
| $\ldots, a_{moveObjContainer}, a_{updateObjState}\}$ $(\mathcal{A_3})$ | ✗ | 0.45 | 0.48 | 0.03 |
| | ✓ | 0.17 | 0.60 | 0.43 |
| $\ldots, a_{moveObjContainer}, a_{moveObjRoom}\}$ $(\mathcal{A_4})$ | ✗ | 0.12 | 0.73 | 0.62 |
| | ✓ | 0.13 | 0.37 | 0.23 |
| $\ldots, a_{moveObjContainer}, a_{info-*}\}$ $(\mathcal{A_5})$ | ✗ | 0.09 | 0.45 | 0.37 |
| | ✓ | 0.10 | 0.42 | 0.32 |
| $\ldots, a_{moveObjContainer}, a_{moveObjRoom}, a_{info-*}\}$ $(\mathcal{A_6})$ | ✗ | 0.08 | 0.31 | 0.23 |
| | ✓ | 0.13 | 0.51 | 0.38 |
| $a_{moveObjContainer}, a_{moveObjRoom}, a_{chitChat-*}, a_{info-*}\}$ $(\mathcal{A_7})$ | ✗ | 0.73 | 0.86 | 0.13 |
| | ✓ | 0.75 | 0.85 | 0.10 |
| $\ldots, a_{chitChat-private}\}$ $(\mathcal{A_8})$ | ✗ | 0.75 | 0.82 | 0.07 |
| | ✓ | 0.68 | 0.76 | 0.08 |
| $\ldots, a_{chitChat-public}\}$ $(\mathcal{A_9})$ | ✗ | 0.61 | 0.68 | 0.07 |
| | ✓ | 0.54 | 0.61 | 0.07 |

Table 5 shows a breakdown across all settings. 40% of the stories were infilled in a single attempt. Next step options are sampled simultaneously with repetition penalty for added wording diversity. This is due to the stringent LLM as a judge conditions to ensure quality. Without these quality and

diversity constraints, 81% of the story structures are infilled within a single attempt (73% with only quality constraints).

### B.3 MODELS FAIL BOTH AT THEORY OF MIND AND PURE STATE TRACKING (CONT. FROM § 5)

Table 6: Accuracy breakdown of the experiment shown in Table 1, discriminating if each question is *interesting* or not. A question is interesting if the answer would change depending on the entity asked about, thus potentially requiring theory of mind. Results show that part of a model's difficulty with EXPLORETOM's generated data could be attributed to poor state tracking (i.e., the uninteresting questions, noted ¬Int.).

| Action Set | Includes Symmetry? | Llama Acc. Int. | Llama Acc. ¬Int. | Llama % Int. | GPT4o Acc. Int. | GPT4o Acc. ¬Int. | GPT4o % Int. | Mixtral Acc. Int. | Mixtral Acc. ¬Int. | Mixtral % Int. |
|---|---|---|---|---|---|---|---|---|---|---|
| $\mathcal{A}_1$ | ✗ | 0.42 | 0.20 | 44% | 0.47 | 0.43 | 45% | 0.44 | 0.38 | 48% |
|  | ✓ | 0.31 | 0.01 | 49% | 0.46 | 0.28 | 49% | 0.37 | 0.40 | 50% |
| $\mathcal{A}_2$ | ✗ | 0.61 | 0.25 | 42% | 0.57 | 0.24 | 41% | 0.39 | 0.01 | 50% |
|  | ✓ | 0.49 | 0.22 | 50% | 0.56 | 0.23 | 49% | 0.19 | 0.00 | 34% |
| $\mathcal{A}_3$ | ✗ | 0.54 | 0.24 | 47% | 0.55 | 0.35 | 47% | 0.46 | 0.18 | 46% |
|  | ✓ | 0.42 | 0.04 | 50% | 0.57 | 0.31 | 48% | 0.36 | 0.04 | 48% |
| $\mathcal{A}_4$ | ✗ | 0.56 | 0.11 | 16% | 0.58 | 0.09 | 25% | 0.49 | 0.00 | 25% |
|  | ✓ | 0.25 | 0.18 | 49% | 0.47 | 0.21 | 35% | 0.44 | 0.00 | 31% |
| $\mathcal{A}_5$ | ✗ | 0.36 | 0.09 | 50% | 0.50 | 0.30 | 45% | 0.44 | 0.40 | 48% |
|  | ✓ | 0.24 | 0.15 | 50% | 0.44 | 0.29 | 48% | 0.44 | 0.47 | 50% |
| $\mathcal{A}_6$ | ✗ | 0.37 | 0.18 | 50% | 0.64 | 0.25 | 44% | 0.50 | 0.03 | 48% |
|  | ✓ | 0.31 | 0.14 | 50% | 0.64 | 0.25 | 41% | 0.49 | 0.04 | 48% |
| $\mathcal{A}_7$ | ✗ | 0.74 | 0.74 | 46% | 0.76 | 0.76 | 44% | 0.58 | 0.72 | 42% |
|  | ✓ | 0.74 | 0.67 | 43% | 0.72 | 0.75 | 47% | 0.47 | 0.67 | 42% |
| $\mathcal{A}_8$ | ✗ | 0.81 | 0.62 | 67% | 0.80 | 0.72 | 66% | 0.55 | 0.42 | 67% |
|  | ✓ | 0.62 | 0.59 | 74% | 0.63 | 0.37 | 75% | 0.41 | 0.42 | 74% |
| $\mathcal{A}_9$ | ✗ | 0.48 | 0.67 | 40% | 0.50 | 0.42 | 40% | 0.52 | 0.27 | 40% |
|  | ✓ | 0.54 | 0.51 | 50% | 0.65 | 0.41 | 50% | 0.54 | 0.27 | 50% |
| Total | — | 0.49 | 0.31 | 48% | 0.58 | 0.37 | 47% | 0.45 | 0.26 | 47% |

### B.4 HOW LIKELY IS A RANDOMLY-SAMPLED STORY TO REQUIRE THEORY OF MIND? (CONT. FROM §5)

Table 7: Probability that a randomly-sampled story would require theory of mind for answering at least one question. Actions considered are $\{a_{\text{enter}}, a_{\text{leave}}, a_{\text{moveObjContainer}}\}$, all settings of $\{2, 3, 4\}$ people and $\{2, 3, 4\}$ $a_{\text{moveObjContainer}}$ movements, with 10 maximum actions, are shown.

| Number of people | Number of movements 2 | 3 | 4 |
|---|---|---|---|
| 2 | 0.131 | 0.208 | 0.235 |
| 3 | 0.195 | 0.234 | 0.288 |
| 4 | 0.210 | 0.259 | 0.315 |

Table 8: Probability that a randomly-sampled (story, question) pair would potentially require theory of mind, meaning that the answer to the question varies depending on the entities considered. Actions considered are $\{a_{\text{enter}}, a_{\text{leave}}, a_{\text{moveObjContainer}}\}$, all settings of $\{2, 3, 4\}$ people and $\{2, 3, 4\}$ $a_{\text{moveObjContainer}}$ movements, with 10 maximum actions, are shown.

|  | Number of movements | | |
|---|---|---|---|
| Number of people | 2 | 3 | 4 |
| 2 | 0.090 | 0.123 | 0.124 |
| 3 | 0.120 | 0.109 | 0.121 |
| 4 | 0.111 | 0.101 | 0.112 |

Table 9: Probability that a randomly-sampled (story, question, answer) triple would require an answer that is different from the true world state (i.e., it is a *false-belief question*. Actions considered are $\{a_{\text{enter}}, a_{\text{leave}}, a_{\text{moveObjContainer}}\}$, all settings of $\{2, 3, 4\}$ people and $\{2, 3, 4\}$ $a_{\text{moveObjContainer}}$ movements, with 10 maximum actions, are shown.

|  | Number of movements | | |
|---|---|---|---|
| Number of people | 2 | 3 | 4 |
| 2 | 0.059 | 0.084 | 0.086 |
| 3 | 0.065 | 0.065 | 0.072 |
| 4 | 0.056 | 0.049 | 0.058 |

### B.5 ON WHY A STORY WITH A GREATER NUMBER OF PEOPLE MAY COUNTERINTUITIVELY IMPLY A LOWER AVERAGE DIFFICULTY

In EXPLORETOM, the number of people and actions is important from a controllability and diversity perspective, but does not directly quantify task difficulty—difficulty quantification for theory of mind is an active area of research. Huang et al. (2024) quantifies a theory of mind problem complexity as the number of states necessary to solve it correctly (note that their approach requires manual annotation). While the number of states tends to increase with the number of people and actions, many questions do not require analyzing the whole story, e.g. if someone only entered the scene right at the end. When randomly sampling stories while fixing the number of core actions to e.g. 5, it's more likely to have some characters with little involvement in the scene if there are 5 people in total than if there are 2 people. Since accuracy is computed across all questions about all characters, having a larger number of people may bump the average accuracy. EXPLORETOM's flexible framework allows for minimizing these cases through modifying the lookahead, but we chose against both doing this or filtering questions to show the performance is low even without these considerations.

## C  PROMPTS USED FOR GENERATING AND VALIDATING EXPLORETOM'S DATA

### C.1  GENERATING STORY CONTEXTS (CONT. FROM §2.1)

```
Suggest a short context where {num_people} people are together in a
room. It should be at most two sentences long, and they should be able
to observe each other. Later in the story, characters are going to move
 around and store objects, so your context should be plausible under
those constraints. Do not explicitly include that they can all see each
 other, it should be clear from context. The room could be in a house,
work environment, etc.

Here's an example for three people. Follow the same format.

LIST CHARACTERS' NAMES:
1. Emily, a meticulous office manager.
2. Jason, a tech-savvy intern.
3. Karen, a diligent accountant.

GIVE SHORT STORY CONTEXT:
Emily, Jason, and Karen gathered around the central table in the sleek
office's conference room, discussing the upcoming audit. As they
strategized, the shelves and storage compartments lining the walls
around them held the tools and documents they would soon need to
organize and pack away.

ROOM IN WHICH THIS STORY BEGINS:

NAME ONE REASONABLE ALTERNATIVE ROOM THEY COULD MOVE TO:

NAME ONE OBJECT TO BE MOVED BY A PERSON DURING THE STORY:

LIST {num_containers} REASONABLE OPAQUE CONTAINERS THAT COULD CONTAIN
THIS OBJECT:

LIST {num_topics} DISTINCT AND REASONABLE TOPICS THEY COULD BE CHATTING
 ABOUT:

To get inspired, make this context happen in {sampled_location}.
Suggested names are {sampled_names}, but feel free to come up with your
 own names if it would suit the story better. Be direct with your
answers: do not include parentheses or clarifications beyond the
responses requested. Do not refer to plural objects or give options if
a singular thing is requested. The object could be anything--an apple,
a pen, a spoon, a pair of scissors, a chocolate bar, etc.--, be
creative! Avoid vases and microphones, or adding too many details to
the object's description in general.
```

Figure 7: Prompts used for generating a story context, after infilling the variables (number of people, containers, topics, names, and location). Names and location are sampled independently to increase diversity, prompts shown in Fig. 8.

```
List 100 names. Do not include any other text.
```

```
Suggest 100 different general contexts in which a story may happen. The
 context should be able to have several people in the same location
easily listening and observing each other.

1. a school
2. a hospital
3. a vet shop
4. a family living room

Follow the format and make the descriptions as short as possible. Do
not include any text before the list.
```

Figure 8: Prompts used for generating a list of possible characters' names and locations for the story.

## C.2  PROMPTS USED FOR STORY INFILLING

```
You are an expert writer that uses simple language, avoiding sounding
unnatural or cliché. You are clear, creative, and helpful. You use
simple sentence constructions and words so that everyone may understand
 you.
```

Figure 9: System prompt used for story infilling

```
Given the following story and knowing the description of the characters
 involved, write the start of a story. Don't actually describe any
actions in the story, just the setting in which the story will happen.
Only include the characters that are mentioned in the story.

STORY:
{story_script}

CHARACTERS:
{characters_description}

TWO-SENTENCE STORY BEGINNING THAT DOES NOT INCLUDE OR SUGGEST ANY
INFORMATION OF WHAT WILL HAPPEN IN THE STORY. DO NOT MENTION PEOPLE:
```

Figure 10: System prompt used for sampling narration (the start of the story, before infilling).

```
Given the following story and knowing the description of the characters
 involved, suggest a reasonable goal for each character. Only include
the characters that were mentioned in the story.

STORY:
{story_script}

CHARACTERS:
{characters_description}

CHARACTERS GOALS: <insert here>

Follow the format and do not include any other text. Only include the
characters mentioned in the story, and do not even mention the others
in your list.
```

Figure 11: System prompt used for sampling character goals.

```
Continue the story {story_length}, clearly conveying the action or
information below without altering it. Do not contradict any prior
information. Avoid repeating the information verbatim, instead
naturally (and possibly implicitly, but still unambiguously) conveying
the meaning. Do not add characters or actions that were not explicitly
described. Do not replace characters even if this would improve flow.
Combining actions into a single sentence is OK as long as you do not
alter the original information. {infilling_text_type}

Make it a short, yet an interesting story to read. Make the text
exciting to read as well as each character's speech, so try to avoid e.
g. starting all the sentences the same way. The story needs to follow
common sense, e.g. do not magically change an object's location without
 mentioning it. Do not include any notes, comments, parentheses, or any
 other form of extra text that would not belong in a story. Feel free
to hint or describe characters' goals and motivations for performing
the actions if it would make the story flow better.

As a warning, take into account that when someone tells someone
privately they might not be in the same location, e.g. they might be
sending a text message or making a phone call; they might also be in
the same location, in that case they could also communicate through a
gesture, a whisper, etc. Do not assume a person is in the same room if
it has not been made explicit before. Also, if someone was spying, or
if they were distracted and did not listen or saw something happen, do
not forget to include it! Remember that in this case, it should be
clear that the distraction or spying applies only to the action
mentioned and they go back to normal after the action is finished.
Avoid making multiple copies of the same object.

Give {num_tries_completions} responses, ensuring to give {
num_tries_completions} different phrasings of continuing the story
conveying the action. Use very different wordings and sentence
structures, but avoid changing object or room names!

WHO ARE THE CHARACTERS: {people_with_personas}

WHAT ARE THEIR GOALS: {optional_characters_goals}

NEW ACTION OR INFORMATION TO INCLUDE: {new_information}

CURRENT SHORT STORY: {story_context}

Follow the format and do not include any other text. Do not include any
 text before the list. Do not enumerate. Continue the story {
story_length}. Avoid repeating the information verbatim, instead
naturally (and possibly implicitly, but still unambiguously) conveying
the meaning.

STORY CONTINUATION: <fill>

STORY CONTINUATION: <fill>
```

Figure 12: Prompt used for iterative story infilling including characters' goals, and allowing for simultaneous sampling of several possible infillings, which when associated with repetition penalty, yields more diverse infillings. Infilling length is uniformly chosen between *'with a single sentence'* and *'with up to two sentences'*, and infilling text type is uniformly chosen between *'Make the new text be declarative, without including conversations.'* and *'Make the new text conversational, using direct quotes to convey the words spoken by a character.'*

# D    EXPLORETOM EXAMPLES (CONT. FROM §3)

See a large sample of EXPLORETOM-generated data in `https://huggingface.co/datasets/facebook/ExploreToM`. We also include a few examples below.

## D.1    STORY STRUCTURE EXAMPLES

---

- Addison entered the monastery dining hall.
- Addison filled the large ceramic vase with fresh sunflowers.
- Addison left the monastery dining hall.
- Charlotte entered the monastery dining hall.
- Charlotte painted the large ceramic vase with intricate designs in gold.
- Charlotte glued a few loose diamonds around the neck of the large ceramic vase. While this action was happening, Addison witnessed this action in secret (and only this action).

---

- Amelia entered the staff room.
- Amelia moved the large first aid kit to the plastic storage bin, which is also located in the staff room. While this action was happening, Alexis witnessed this action in secret (and only this action).
- Amelia entered the equipment storage room.
- Amelia left the equipment storage room.
- Amelia entered the staff room.
- Amelia moved the large first aid kit to the equipment storage room, leaving the plastic storage bin in its original location.
- Amelia moved the large first aid kit to the metal cabinet, which is also located in the equipment storage room.

---

- Alexander entered the city hall planning department.
- Alexander moved the large map of the city to the cardboard tube, which is also located in the city hall planning department.
- Alexander told privately to Leslie that the large map of the city is in the cardboard tube.
- Victoria entered the city hall planning department. Leslie told privately to Peyton that the large map of the city is in the cardboard tube.
- Victoria moved the large map of the city to the plastic storage bin, which is also located in the city hall planning department.

---

Figure 13: Examples of the following story structures: $a_{\text{updateObjState}}$ ($\mathcal{A}_2$) with asymmetry; $\{a_{\text{moveObjContainer}}, a_{\text{moveObjRoom}}\}$ ($\mathcal{A}_4$) with asymmetry; and $\{a_{\text{moveObjContainer}, a_{\text{info}}}\}$ ($\mathcal{A}_5$) without asymmetry.

## D.2 STORY INFILLING EXAMPLES

Table 10: Example of adversarially sampled story structure and iterative infilling with Llama-3.1 70B Instruct, including a story context (first row). This example showcases how the infilling process is able to group sentences when they depict the same event but for more than one person, allowing for an even more natural-sounding story.

| Story Structure (list of actions) | Infilling for each action |
|---|---|
| \<initial context\> | The operations center, a bustling hub of activity during the park's peak festival season, was dimly lit except for a few overhead fluorescent lights that cast a sterile glow over the rows of workstations and storage units. The air was thick with the smell of stale coffee and fresh printer paper, a familiar scent to anyone who spent time behind the scenes of the busy amusement park. |
| Brooklyn entered the operations center.
James entered the operations center. | As she entered the operations center, Brooklyn's presence was swiftly followed by James, whose rugged demeanor stood in contrast to the sterile environment of the room. |
| James moved the large festival schedule poster to the cardboard box, which is also located in the operations center. | James neatly relocated the oversized festival schedule poster, carefully placing it within the confines of a cardboard box, a receptacle that had been securely positioned in a corner of the operations center. |
| James left the operations center.
Brooklyn left the operations center. | Brooklyn stepped out into the bright sunlight, a stark contrast to the dimly lit operations center she had just left, while James followed suit, the warm rays illuminating his weathered features. The festival preparations were now unfolding across the park, with both of them poised to tackle their respective tasks. |
| Tyler entered the operations center.
Brooklyn entered the operations center. | As Tyler walked into the operations center, the soft hum of idle machinery and the faint scent of printer paper enveloped him, a familiar yet calming atmosphere before the chaos of the festival. The room's sterile lighting seemed to intensify, reflecting off the polished surfaces as Brooklyn slipped in, her footsteps quiet on the worn floor. |
| Brooklyn moved the large festival schedule poster to the plastic storage bin, which is also located in the operations center. | Brooklyn's hands moved swiftly as she carefully slid the oversized festival schedule poster into the large plastic storage bin in the operations center. |

Table 11: Example of adversarially sampled story structure and simple iterative infilling with Llama-3.1 70B Instruct, along with example questions and model answers.

| Story Structure (list of actions) | Infilling for each action |
|---|---|
| David entered the study room. Sarah entered the study room. | David walked into the study room, scanning the organized chaos of files on the desk, just as Sarah stepped in, clutching her tablet and a determined look. |
| Sarah moved the prototype model to the metal filing cabinet, which is also located in the study room. | Sarah moved the prototype model to the metal filing cabinet, her fingers brushing against the cool surface before turning to David with a confident smile. |
| David left the study room. | After a brief exchange with Sarah, David grabbed his briefcase and left the study room, mindful of the upcoming meeting downtown. |
| Mark entered the study room. | Mark entered the study room, his eyes quickly assessing the scattered documents and half-finished sketches on the table, before settling into his favorite chair by the window to brainstorm new ideas. |
| Mark moved the prototype model to the wooden chest, which is also located in the study room. | Mark, contemplating his latest concept, carefully picked up the prototype model from the metal filing cabinet and transferred it to the wooden chest, a better place for such an inspiration-spurring creation. |

| Selected Questions | EXPLORETOM -generated answer using our domain-specific language | GPT-4o's responses | Correct? |
|---|---|---|---|
| In which container will David search for the prototype model? | metal filing cabinet | david will search for the prototype model in the metal filing cabinet. | Yes |
| In which container will Sarah search for the prototype model? | wooden chest | sarah will search for the prototype model in the metal filing cabinet. | No |
| In which container will Mark search for the prototype model? | wooden chest | mark will search for the prototype model in the wooden chest. | Yes |
| In which container will David search for the prototype model? | metal filing cabinet | the metal filing cabinet. | Yes |
| In which container will Sarah search for the prototype model? | wooden chest | sarah will search for the prototype model in the metal filing cabinet. | No |
| In which container will Mark search for the prototype model? | wooden chest | the wooden chest. | Yes |

