# OpenReview forum: "Explore Theory of Mind: program-guided adversarial data generation for theory of mind reasoning"
_ICLR.cc/2025/Conference — ICLR 2025 Poster_

### Official Review · Reviewer_p4tC · 2024-11-02

**Soundness:** 3
**Presentation:** 2
**Contribution:** 3
**Rating:** 6
**Confidence:** 3

**Summary:**

This paper introduces the TrackTheMind method, which is used to generate a theory of mind story with specific constraints, such as having exactly 3 people in the story.

Generally speaking, TrackTheMind is a tree search process. It starts from a "root node": TrackTheMind uses an LLM to generate a context, including characters, environment, objects, and possible actions. Then, it generates n leaf nodes from this node, where each leaf node can contain n actions that modify the environment state. Among these n leaf nodes, A search is used to select one while discarding the others. The A value function f(s) = g(s) + h(s), where g(s) is the accuracy rate of all questions that the LLM can generate at leaf node s, and h(s) is the probability that subsequent nodes from this leaf node can fulfill the specific constraints.

The authors first used TrackTheMind to generate evaluation data, demonstrating that current LLMs still need improvement in their performance on complex theory of mind datasets. Furthermore, the authors used TrackTheMind to generate training data, and experimental results showed that this training data can effectively improve the model's theory of mind capabilities while maintaining the model's basic utility.

**Strengths:**

1. This paper is well-structured and generally easy to follow, except for Section 2, especially in describing the overall TrackTheMind pipeline and the description of the A* search.
2. The types of ToM questions considered are comprehensive, especially those containing asymmetric belief updates, which can create complex questions.
3. The ToM question generation process is automatic, and given the 'tree' structure, its correctness can be easily verified.

**Weaknesses:**

1. First, how can we quantitatively evaluate the complexity of the generated ToM stories? If complexity is quantified by the number of people and actions involved, why do the experiments in Fig 3 show that model performance increases as the number of people involved increases?
2. In A* search, g(s) requires to evaluate LLM performance of the entire question generated by state s, which maybe time-consuming.
3. The authors demonstrated that models trained on the TrackTheMind training set largely maintain their utility. However, only Multi3Woz and MMLU were evaluated. I expect to evaluate it on more common datasets as it is easy to implement.
4. In Section 2.1, the story context structure is simple and may not be general enough for complex, real-world scenarios.

**Questions:**

Please refer to the weaknesses part.

---

> ### Author Response · Authors · 2024-11-21
> **Response to Reviewer p4tC**
>
> We thank the reviewer for their feedback! We're glad that they mentioned that TrackTheMind's structured generation design enables asking questions whose "correctness can be easily verified", and that the types of ToM questions considered are "comprehensive".
>
> We’d like to emphasize that on top of being able to create a useful training data set and a challenging evaluation set tailored to any particular model, TrackTheMind enables performing conceptual analyses, like 1. quantifying how the balance between interesting and uninteresting ToM data seriously affects downstream performance and 2. analyzing model performance in underlying skills for false-belief tasks such as state tracking (see Section 5).
>
> **We address all questions below, and have added several new analyses in the general response**, including o1-preview evaluation, a detailed cost analysis, and detailed dataset statistics.
>
> ---
> ### _“In Section 2.1, the story context structure is simple and may not be general enough for complex, real-world scenarios.”_
> We agree, and we mention this in the Limitations section. We would like to however emphasize that TrackTheMind still represents a radical improvement in richness and diversity with respect to prior work! We will release our code to enable the community to add support to other action sets. In the future we envision that we could even generate code for each action with LLM assistance.
>
> ---
> ### _“How can we quantitatively evaluate the complexity of the generated ToM stories? If complexity is quantified by the number of people and actions involved, why do the experiments in Fig 3 show that model performance increases as the number of people involved increases?”_
> Please see general response for a detailed explanation.
>
> ---
> ### _On the cost of A* search_
> Running A* with a budget of N nodes (i.e., partial stories) costs exactly N times the cost of evaluating that node (story) if it were to be included as part of a final dataset. If efficiency is a concern, then N can be modified as desired. However, we believe that for the evaluation benchmark application it is crucial to focus on finding the most challenging data points for evaluation, and speed is not as central, hence why we selected N=50, which totals an average cost of less than $0.05 per TrackTheMind story generated for GPT-4o. **See a detailed cost analysis in the general response!**
>
> ---
> ### _“The authors demonstrated that models trained on the TrackTheMind training set largely maintain their utility. However, only Multi3Woz and MMLU were evaluated. I expect to evaluate it on more common datasets”_
> **We evaluated on MMLU since we believe that it is one of the most widely used reasoning benchmarks**, and would like to emphasize that **this analysis was simply intended as a confirmation that general reasoning capabilities did not severely degrade**, and thus it would not be impractical to add TrackTheMind to a more general SFT stage. Other works have used MMLU with this goal (e.g. Wang et al., 2024). We are happy to evaluate on other benchmarks too!
>
> ---
> ### _“TrackTheMind is a tree search process. [...] TrackTheMind generates n leaf nodes from this node [...] Among these n leaf nodes, A search is used to select one while discarding the others.”_
> We would like to clarify that this is not exactly what we do: **we follow the classic A* search algorithm**. A* uses a priority queue to decide what node $s$ to explore next based on their value $f(s)=g(s)+h(s)$. Our contribution is in designing $g(s)$ and $h(s)$, and in making A* feasible by not exploring all possible next nodes. We will make sure to emphasize this even more in Section 2.
>
> ---
> We hope this solves all the reviewer’s questions and we are happy to discuss any remaining concerns!
>
> References
> - Wang, Ruiyi, et al. "SOTOPIA-$\pi $: Interactive Learning of Socially Intelligent Language Agents." 2024.

---

> > ### Comment · Reviewer_p4tC · 2024-11-24
> > **Thank you for your response**
> >
> > I appreciate the author's thorough response. It addressed most of my concerns. I will keep my positive score at 6 as there is no 7 option.

---

> > > ### Author Response · Authors · 2024-11-26
> > > **Thanks for your consideration**
> > >
> > > Thanks for considering '7' for our paper even if the option is not available! We are happy to address any other questions or suggestions you might have.

---

### Official Review · Reviewer_XWk4 · 2024-11-03

**Soundness:** 3
**Presentation:** 3
**Contribution:** 2
**Rating:** 6
**Confidence:** 4

**Summary:**

The paper introduces TrackTheMind, a framework for generating challenging theory of mind (ToM) testing and training data for LLMs. To generate stories, this work samples plausible contexts, uses A* search to find challenging story structures, and infills these with an LLM. The results show that LLMs seriously struggle on some scenarios, potentially due to poor state tracking skills and the scarcity of training data that specifically requires ToM reasoning, which can be alleviated to some degree by finetuning.

**Strengths:**

1- Do large language models (LLMs) have theory of mind? I think this is a very important research question!
2- Overall, the paper does a good job of presenting arguments and claims.
3- The proposed benchmark seems to be very challenging for LLMs, as indicated by the results.

**Weaknesses:**

1- The paper argues that "basic theory of mind is still elusive to LLMs," and thus this "demonstrates the need for training data that purposefully requires theory of mind." Do the authors think the lack of theory of mind skills can be "resolved" (we know it can be "alleviated") with enough training data? The results on the FANToM benchmark in Table 3 suggest that even finetuning on 114,000 data points of TrackTheMind does not necessarily improve the theory of mind abilities of LLMs. Instead, the reported gain can be explained by the fact that the proposed benchmark is similar to some benchmarks, and by training on TrackTheMind data, models can perform better on similar benchmarks like ToMi without really developing an internal skill that can be generalized across other scenarios.

2- While providing a new benchmark is a contribution, in terms of "new insights," it is not very clear to me how much contribution this work makes. Several other works are suggesting the lack of abilities in the context of theory of mind. But it is not clear to me what "new" insights this work offers researchers that cannot be obtained from other similar works.

While I appreciate the effort for development of this new and challenging benchmark, the work falls short of providing novel insights into theory of mind capabilities in LLMs.

**Questions:**

In addition to the questions above, I have the following question:

1- In the caption of Figure 3, the authors mention that "A story with greater number of people suggests lower difficulty, possibly because there is a fixed number of actions, thus fewer actions per person." However, I'm not sure if I completely followed the reasoning here. When representing the state programmatically, we need to include the status of each person before/after action. So I would argue the number of people has an impact on the state size, and also total number of actions has an impact on number of times we need to update state. Thus, both of them should have an impact on difficulty, but Figure 3 shows otherwise. Could the authors explain this?

---

> ### Author Response · Authors · 2024-11-21
> **Response to Reviewer XWk4, Part 1**
>
> We thank the reviewer for their feedback. We are glad they mentioned that we explore a “very important research question”, and we address their concerns below. **We have also included a new report of dataset statistics, extra cost analyses and model evaluations, plus context sampling details in the general response.**
>
> ---
> ### _“Benchmark seems to be very challenging for LLMs” [...] “But is not clear to me what "new" insights this work offers”._
> Thanks for giving us the opportunity to clarify this. TrackTheMind is a synthetic data generation framework to alleviate the issue of lack of naturally occurring data—it can then be used as a benchmark, but it also enables research on training for developing ToM (see next answer for details).
>
> TrackTheMind is the first to enable stress-testing a model by generating reliable stories targeted to it. This makes our framework more robust to leakages–-a problem that has made human-generated benchmarks difficult to use over the years—and the issue of accidentally developing a ToM benchmark that is already close to saturation.
>
> **When TrackTheMind is used as training data, it not only increases benchmark performance, but also generalizes better than previously possible** (e.g. see Sclar et al., 2023’s fine-tuning on ToMi).
>
> **TrackTheMind enables quantifying how the balance between interesting and uninteresting ToM data seriously affects downstream performance** (Figure 5); this gives suggestions for data balancing during SFT for future work. **Our work also enables analyzing underlying skills for false-belief tasks such as state tracking, whose unexpected results continue to challenge the notion that humans and LLMs may learn this skill in similar ways**, as reviewer rkK5 insightfully points out.
>
> Besides these conceptual insights, TrackTheMind radically improves story structures’ richness and naturalness with respect to prior work, giving a first step towards exploring the generation of controlled, more natural scenarios.
>
> Please see the intro for more details, and we’re happy to discuss more!
>
> ---
> ### _“Several other works are suggesting the lack of abilities in the context of theory of mind.”_
> True, but other recent works (e.g., Kozinski 2023, Street et al., 2024) argue the opposite, by possibly considering easier or leaked stories. Thus, having a data generation framework to stress-test ToM skills becomes vital to combat hype in current and future models!
>
> ---
> ### _“Do the authors think the lack of theory of mind skills can be "resolved" (we know it can be "alleviated") with enough training data?”_
> Great question! To the best of our knowledge, prior work does not even explore the question of *alleviating* the lack of ToM skills—this is part of our motivation. We aim to move away from the current status quo in LLM + ToM research that solely measures ToM capabilities, and instead provide a framework to generate data that can also be used in training.
> We believe that it is unclear whether **_any_** reasoning skill can be fully “resolved” in LLMs. For example, Dziri et al., 2023 showed that massive fine-tuning models fails to teach models to solve 5-digit multiplication, and later Deng et al., 2024 developed a training technique that enables models to perform 12-digit multiplication—and still eventually performance decays. One of our goals with TrackTheMind is to enable future research in training models that can develop theory of mind skills by providing the basic, high quality building blocks (i.e., data!).
>
> [response continues in the following message]

---

> ### Author Response · Authors · 2024-11-21
> **Response to Reviewer XWk4, Part 2**
>
> ---
> ### _“The results on the FANToM benchmark in Table 3 suggest that even finetuning on 114,000 data points of TrackTheMind does not necessarily improve the theory of mind abilities of LLMs”_
> We believe that FANToM’s performance is possibly correlated with the fact that this benchmark requires reasoning over a significantly longer context (1000+ tokens) than our current training data: although TrackTheMind is flexible and could generate stories of any length, during infilling we request to use up to 2 sentences. This length confounder has been mentioned in other works (e.g. Llama 3 report 4.3.4.) where they carefully balance data length. Moreover, FANToM authors report that even when fine-tuning on their own data they were not able to make progress on FANToM. This might point to an underlying issue in transformers, or may be a data scale issue given the longer context of this dataset. ToM scaling laws would be exciting to investigate in future work!
>
> ---
> ### _“[...] reported gain can be explained by the fact that the proposed benchmark is similar to some benchmarks [...]”_
> While we still have a long way to go to imbue models with general theory of mind (we definitely don’t claim to have solved this area of research!), BigToM and OpenToM are false-belief datasets that involve situations and questions not covered in TrackTheMind (e.g. OpenToM often asks about feelings). Also, HiToM involves questions of a higher-order ToM than what we currently support in TrackTheMind. We actually produce data that has limited lexical similarity (high phenomenological similarity) to the comparison benchmarks specifically so that we can assess the impact of exposure to the reasoning patterns rather than lexical similarity or other superficial features that might benefit performance without tackling the underlying ToM research questions.
>
> ---
> ### _On why a story with a greater number of people may not necessarily correlate with higher difficulty, and why # people and # actions is not by itself a complexity metric._
> Please see general response for a detailed explanation.
>
> ---
> We hope this solves all the reviewer’s questions and we are happy to discuss any remaining concerns!
>
> References
>
> - Dziri, Nouha, et al. Faith and Fate: Limits of Transformers in Compositionality. 2023.
> - Deng, Yuntian, et al. From explicit cot to implicit cot: Learning to internalize cot step by step. 2024.
> - Huang, X. Angelo, et al. A Notion of Complexity for Theory of Mind via Discrete World Models. 2024.
> - Kosinski, Michal. Theory of mind might have spontaneously emerged in large language models. 2023.
> - Street, Winnie et al. LLMs achieve adult human performance on higher-order theory of mind tasks. 2024.

---

> ### Author Response · Authors · 2024-11-25
> **Discussion period is about to end**
>
> Dear reviewer,
>
> Thank you so much for your original review. The discussion period ends tomorrow: we would really appreciate to hear your thoughts regarding our answers to your questions and the new experiments we included in our response!
>
> Thanks a lot,
> TrackTheMind's authors

---

> ### Author Response · Authors · 2024-12-02
> **Extended discussion period is about to end**
>
> Dear reviewer,
>
> Thank you for taking the time to provide your original review! The extended discussion period ends tomorrow: we would really appreciate to hear your thoughts on our response. We believe we've addressed your questions, but would love to have the chance to address any remaining ones!
>
> Thanks again,
>
> TrackTheMind's authors

---

> ### Comment · Reviewer_XWk4 · 2024-12-02
>
> I would like to thank the authors for their response to my questions. While I still the paper is borderline, I believe the new revision has noticeable improvements and the strengths outweigh the weaknesses. Thus, I would increase my score to 6.

---

### Official Review · Reviewer_BPYe · 2024-11-03

**Soundness:** 3
**Presentation:** 4
**Contribution:** 3
**Rating:** 6
**Confidence:** 4

**Summary:**

This paper introduces a novel methodology for generating program-based Theory of Mind stories using state control, difficulty evaluation, and A* search to ensure a suitably challenging benchmark. The authors conduct two main experiments:

- Benchmark Evaluation: They first evaluate the performance of LLMs on the new benchmark created with TrackTheMind-generated stories. Results indicate that even advanced models struggle with this benchmark, highlighting its potential as a rigorous test for mind reasoning.

- Model Fine-Tuning with Synthesized Data: Using their framework, the authors synthesize training data to fine-tune a model, resulting in significant improvements on both in-domain and out-of-domain benchmarks.

Additionally, the authors offer insights into potential factors contributing to the observed limitations in model performance on mind reasoning tasks.

**Strengths:**

- A novel framework for synthesizing mind reasoning data.
- A sufficiently challenging benchmark to evaluate the mind reasoning capabilities of LLMs.
- A robust training set offering more data, a complex structure, and strong generalization potential.
- Facilitates investigation into why mind reasoning tasks remain challenging for LLMs.

**Weaknesses:**

- The predefined action sets may limit the variety and richness of the story, potentially constraining creativity and depth.
- Other weaknesses align with the questions section, where I have shared thoughts on things needing further explanation.

**Questions:**

- In the title, you mention "adversarial," but there is little explicit explanation of what makes the dataset adversarial. Could you expand on this concept?
- Could you provide additional statistics on your synthetic dataset to offer a clearer understanding of its characteristics? I think detailed dataset statistics are often essential in synthetic data-related research.
- Is there a significant difference in the quality of synthetic stories generated by different models, such as Llama3-8B-Instruct and Llama3-70B? It would be useful to investigate how the varying capabilities of these models impact the quality and characteristics of the synthetic data.
- If time permits, could you try gathering training data from other Mind Reasoning Datasets to train the Llama3-8B-Instruct model and evaluate it on your benchmark? This cross-evaluation could offer valuable insights into model performance across datasets.

---

> ### Author Response · Authors · 2024-11-21
> **Response to Reviewer BPYe**
>
> We thank the reviewer for their thoughtful review! We are encouraged that the reviewer found that our framework creates a “robust training set” with “strong generalization potential”, that can be also used as a “challenging benchmark to evaluate the mind reasoning capabilities of LLMs” while “facilitating investigation into why mind reasoning tasks remain challenging”. We address all comments below **including a new report of dataset statistics as suggested by the reviewer, and extra cost analyses and model evaluations, plus context sampling details in the general response**. We believe that this resolves all the reviewers’ questions, and we are happy to discuss any remaining ones!
>
> ---
> ### _“The predefined action sets may limit the variety and richness of the story, potentially constraining creativity and depth.”_
> We would like to emphasize that we radically expand the story structures’ richness from prior work: 1. by enabling the automatic sampling of actions that are equivalent to other w.r.t. mental state updates, 2. by adding asymmetry, and 3. by allowing for objects to be moved between rooms. To exemplify #1, “poisoning an apple” is equivalent to “moving an apple from a drawer to the fridge”, in the sense that both updates are visible only to the witnesses, and yet prior work only allowed object movements (see Section 2.2.1). Other examples are “leaving a note inside a book with invisible ink”, “subtly unscrewing a screw from a bike”, etc. **This radical expansion in diversity while still maintaining controllability is a unique addition of TrackTheMind.**
>
> ---
> ### _Could you provide additional statistics on your synthetic dataset?_
> Of course! See general response for details; we are also happy to include additional metrics.
>
> ---
> ### _“If time permits, could you try gathering training data from other Mind Reasoning Datasets to train the Llama3-8B-Instruct model and evaluate it on your benchmark?”_
> We expect this to perform very poorly given the reports from prior work: Sclar et al., 2023 fine-tuned GPT3 with ToMi and showed that it failed to solve even slightly different story structures. The structures analyzed by Sclar et al., 2023 can be generated using TrackTheMind. We will however include this analysis for the camera ready!
>
> ---
> ### _Why is the dataset adversarial?_
> TrackTheMind is a framework designed to search for the most difficult stories for a given model. We call this adversarial since it is stress-testing a specific model for ToM. This is also why we don’t present TrackTheMind as a _dataset_, but rather a _synthetic data generation framework_.
>
> ---
> ### _Is there a significant difference in the quality of synthetic stories generated by different models, such as Llama3-8B-Instruct and Llama3-70B?_
> Story structures are governed by our pre-programmed actions that guarantee the reliability of the predicted mental states regardless of the underlying model. We would however recommend using a strong model for sampling the equivalent actions, as they rely on having a high quality LLM-as-a-judge model—like most synthetic data generation works.
>
> ---

---

> ### Author Response · Authors · 2024-11-25
> **Discussion period is about to end**
>
> Dear reviewer,
>
> Thank you so much for your original review. The discussion period ends tomorrow: we would really appreciate to hear your thoughts regarding our answers to your questions and the new experiments we included in our response!
>
> Thanks a lot,
> TrackTheMind's authors

---

> > ### Comment · Reviewer_BPYe · 2024-12-02
> >
> > Thank you for your response! I think the comment addresses my concern.

---

### Official Review · Reviewer_rkK5 · 2024-11-04

**Soundness:** 2
**Presentation:** 2
**Contribution:** 3
**Rating:** 6
**Confidence:** 3

**Summary:**

This paper proposes TracktheMind, an adversarial data generation pipeline to collect challenging ToM data via A* search.

With adversarial control on the difficulty of the generated data, the collected evaluation data poses a significant challenge to existing LLMs.

The authors also demonstrate the effectiveness of the TracktheMind-generated data as a training corpus to enhance ToM reasoning.

**Strengths:**

* **A controllable and generalizable data generation pipeline to collect ToM reasoning data**.
With predefined ToM-specific language and a rule-based state tracker, the proposed pipeline can automatically collect ToM data of various difficulty levels with high-precision annotated labels.

* **Intriguing results regarding the effect of interestingness in training and evaluation data**.
The superior model performance on interesting questions against the "uninteresting" ones is unexpected and insightful. This may indicate a mechanism different from that of humans in LLMs to tackle ToM tasks.

* **Details of hyperparameter settings and prompt designs**.
The authors provide plenty of details about the hyperparameters and categories of actions, scenarios, etc. they consider in data construction. This ensures the reproducibility and the convincingness of the experimental results in the paper.

**Weaknesses:**

* **Potential bias in topics, scenarios, and stories generated by LLMs**.
The LLMs are included in several crucial stages of the TracktheMind pipeline. For example, the plausible context creation and sampling is important as an initial stage to determine the topics and possible actions that can be covered in the data. However, this process is done by LLMs themselves, which can introduce inherent bias that hinders the generalizability of the generated data. The authors could provide more statistics and strategies they utilize to balance the story topics and scenarios in data generation to better fit real-world situations.

* **Lack of detailed discussion on the exact cost of data generation via A\* search**.
A\* search can be computationally expensive as the size of the search space increases. The authors mentioned that they reduced the cost by restricting the number of neighbors to consider in $f(x)$ evaluation. The authors could elaborate on how this hyperparameter balances the quality, diversity, and cost of data generation and clarify the exact cost (e.g., #tokens) required in different settings. This could help estimate the proposed method's efficiency and how it would work in practice.

* **Lack of deep analysis to disentangle the specific factors that bottleneck the LLM ability of ToM reasoning**.
The results of ablation on #people and #actions in Figure 3 are a bit confusing. On the one hand, the number of actions seems to matter as fewer actions per person reduce the task difficulty. On the other hand, the increase in the number of actions makes little difference in the model performance in the right plot. Unless the variance in performance causes this, given the limited ranges of #people and #actions or number of test samples considered, there might be some factors (or even spurious features) that dominate the model performance. For example, the number of people and actions may not be directly related to the reasoning steps required to answer some ToM questions, whether it is interesting or not. The authors could provide some meso-analysis on the factors that can reflect the task difficulty more directly.

**Questions:**

* As the accuracy of LLMs on some TracktheMind data is quite low (e.g., $5$%), have you tried finer-grained metrics to assess the model's ability? For example, instead of directly enforcing the model to answer yes/no, it would help to diagnose its understanding of the context by extracting its confidence regarding the question and probing the level of uncertainty in the corresponding scenario.

* How the *important actions* are defined to determine a desired user condition? Is this a crucial design to control the generated data's difficulty, quality, and diversity? Would it generalize across different scenarios?

* What is the background of the annotators? Does this matter for the performance in task completion?

* Could you elaborate on the difference among the chosen ToM benchmarks in Table 3? Why the last two did not benefit from the TracktheMind training?

* Why does the model performance on ToMi drop significantly (compared to llama3.1-8b-instruct baseline) when training with 0% of interesting questions? It should be at least the same level as the baseline performance unless I missed something.

* It appears that interestingness and asymmetry are not the crucial factors that impact task difficulty or model performance in evaluation. What might be the cause of such misalignment/inconsistency?

* OpenAI o1 with inference-time scaling may boost the performance by exploring more possibilities for better state tracking. It would provide some insights by assessing it using the TracktheMind-generated ToM data to check whether it can improve performance as expected. This could help to better understand the bottleneck in existing LLMs to tackle such ToM reasoning tasks.

---

> ### Author Response · Authors · 2024-11-21
> **Response to Reviewer rkK5, Part 1**
>
> We thank the reviewer for their thorough feedback! We are pleased that they pointed out that TrackTheMind is a “controllable and generalizable data generation pipeline” with “high-precision annotated labels”, and whose collected data can be used as a “significant challenge to existing LLMs” or as a training corpus. We’re particularly encouraged that the reviewer finds some of the analyses that TrackTheMind enables to be “unexpected and insightful” and may “indicate a mechanism different from that of humans in LLMs to tackle ToM tasks”.
>
> **We address all questions below, including several new analyses and extra information based on the reviewer’s suggestions.** New content includes a breakdown of costs in terms of # tokens used, an analysis using O1-preview, additional insights into our methodology for sampling diverse topics & scenarios, as well as general dataset statistics. We will incorporate all feedback into the camera ready, and we are happy to discuss any remaining concerns!
>
> ---
> ### _Could you analyze OpenAI o1 since inference-time scaling may boost performance by exploring more possibilities for better state tracking?_
> We ran this analysis, please see the general response! We would like to emphasize that **o1-preview was released around the ICLR deadline so it was materially impossible to include it in our original review** (its support is still quite limited), but we also shared the reviewer’s curiosity!
>
> ---
> ### _"Why the last two [benchmarks in Table 3] did not benefit from the TracktheMind training?”_
> OpenToM did show some small gains (F1 score increased from 0.39 to 0.42 (note this is F1, not accuracy), FANToM’s performance is likely affected by a huge length confounder factor (FANToM stories exceed 1000 tokens). In our design, we instruct the model to infill each action using up to 2 sentences, which likely hindered performance.
>
> ---
> ### _“Why does the model performance on ToMi drop significantly when training with 0% of interesting questions? It should be at least the same level as the baseline”_
> Great question! This is because the model adopts a heuristic of relying on ground truth answers, rather than reasoning about mental states when trained without challenging ToM data. **One of our key takeaways is that we need challenging ToM scenarios (and not just state tracking) if we want this kind of reasoning to improve, but unfortunately it is hard to find data that foster deeper reasoning capabilities in the wild—hence the need for TrackTheMind.**
>
> ---
> ### _[Potential bias in topics, scenarios, and stories generated by LLMs] "The authors could provide more statistics and strategies they utilize to balance the story topics and scenarios in data generation to better fit real-world situations”_
> To improve diversity we independently sampled a list of names and scenarios (see Fig. 8) and randomly choose from those before jointly sampling objects, containers, and discussion topics for plausibility. This is a significant improvement from prior work that independently sampled all variables resulting in commonsense violations (see Section 2.1). Any synthetic data generation procedure will introduce some LLM-specific bias, which does not necessarily imply less creative generations (e.g. Chu et al., 2024). Moreover, TrackTheMind could be easily adapted to have human-generated settings as input. Besides this, since we randomly sample the next actions from a defined set, a minimum level of story diversity is ensured as stories will reflect varied combinations from this set (also see dataset statistics in the general response).
>
> ---
> ### _"How the important actions are defined to determine a desired user condition? Is this a crucial design to control the generated data's difficulty, quality, and diversity?"_
> See Experimental Setup (line 318), “[important actions] are the actions that add new basic world knowledge”. This is just to ensure that when a user requests $a$ actions, these are meaningfully advancing a plot, as opposed to e.g. entering and leaving a room immediately after. This is a general definition across scenarios.
>
> ---
> ### _Lack of detailed discussion on the exact cost of data generation via A* search._
> See Experimental Setup, “each story generation is allowed to evaluate 50 nodes” (line 315), which provides a strict upper bound to the cost of generating each data point. **We also added a detailed cost analysis in terms of # tokens as suggested by the reviewer**, please refer to the general response for details!
>
> [continues in the following message!]

---

> > ### Author Response · Authors · 2024-11-21
> > **Response to Reviewer rkK5, Part 2**
> >
> > ---
> > ### _"The results of ablation on #people and #actions in Figure 3 are a bit confusing."_
> > Please see the general response for a detailed explanation.
> >
> > ---
> > ### _“What is the background of the annotators? Does this matter for the performance in task completion?”_
> > We used grad-level educated volunteers, informed by previous works like FANToM, which report that annotator quality suffers (20+ accuracy points of difference!) when using AMT.
> >
> > ---
> > ### _"As the accuracy of LLMs on some TracktheMind data is quite low (e.g., 5%), have you tried finer-grained metrics to assess the model's ability? e.g. probing the level of uncertainty."_
> > We believe that TrackTheMind’s ability to generate short stories that are incredibly challenging is one of its strong points, especially given the current evaluation landscape! While an uncertainty analysis won’t be possible in closed-source models, it would be an interesting follow-up work as models improve performance! (as in e.g. Schaeffer et al., 2024).
> >
> > ---
> > ### _“interestingness and asymmetry are not the crucial factors that impact task difficulty [...]. What might be the cause of such misalignment/inconsistency?”_
> > We do not believe this is not a misalignment: pure state tracking is still elusive for LLMs (the “uninteresting” questions), as well as ToM for all settings (symmetric or otherwise). ToM task difficulty prediction is still an area of research (e.g. Huang et al., 2024), and like most challenging NLP tasks, requires combining multiple skills to solve a task correctly. Our results, as the reviewer insightfully mentions, may point to different mechanisms in LLMs and humans.
> >
> > ---
> > ### _"The authors could provide some meso-analysis on the factors that can reflect the task difficulty more directly."_
> > This is an active area of research in theory of mind + LLM task, e.g. Huang et al., 2024 is concurrent work that focuses solely on measuring this but requires significant labor in manual annotation. We also want to clarify that we do not claim # people and # actions to be the main modulators of complexity!
> >
> > ----
> > We hope this addresses all the reviewers' concerns and we look forward to their response!
> >
> > References
> > - Schaeffer, Rylan, et al. Are Emergent Abilities of Large Language Models a Mirage?. 2024
> > - Huang, X. Angelo, et al. A Notion of Complexity for Theory of Mind via Discrete World Models. EMNLP 2024.
> > - Chu, Junyi, et al. The Task Task: Creative problem generation in humans and language models. 2024

---

> ### Author Response · Authors · 2024-11-25
> **Discussion period is about to end**
>
> Dear reviewer,
>
> Thank you so much for your original review. The discussion period ends tomorrow: we would really appreciate to hear your thoughts regarding our answers to your questions and the new experiments we included in our response!
>
> Thanks a lot,
> TrackTheMind's authors

---

> > ### Comment · Reviewer_rkK5 · 2024-11-25
> >
> > Thanks for the thorough response!
> >
> > > We also want to clarify that we do not claim # people and # actions to be the main modulators of complexity
> >
> > >  FANToM’s performance is likely affected by a huge length confounder factor (FANToM stories exceed 1000 tokens). In our design, we instruct the model to infill each action using up to 2 sentences, which likely hindered performance
> >
> > > We do not believe this is a misalignment ... Our results may point to different mechanisms in LLMs and humans
> >
> > I'd encourage the authors to include an extended discussion of the above challenges in assessing ToM reasoning for existing LLMs to enhance clarity.
> >
> > > When training with 0% of interesting questions, the model adopts a heuristic of relying on ground truth answers, rather than reasoning about mental states when trained without challenging ToM data
> >
> > Thanks for clarifying it! This indeed highlights the advantage of TrackTheMind. You could mention this in the final version to emphasize these findings.
> >
> > > We found that this extensive budget was not enough compute for o1-preview to produce any final answer in 46% of the questions. o1-preview was correct also for 46% of the questions; and plain incorrect for the remaining 8%.
> >
> > I found it unclear how much improvement o1-preview could bring with inference-time scaling. Could the authors report the performance of other models (e.g., GPT-4o) on the same subset (5% of the data analyzed in Table 2) to make a fair comparison?
> >
> > Overall, I appreciate the extended experiment and discussion the authors conducted during the rebuttal. I would correspondingly adjust my score to recommend the acceptance.

---

> > > ### Author Response · Authors · 2024-11-26
> > > **We thank the reviewer and add more details for o1-preview's evaluation**
> > >
> > > We will definitely include an extended discussion and emphasize better the strength points in the intro! Thank you for bringing these points to our attention.
> > >
> > > o1-preview with a budget of 1000 completion tokens obtained accuracy 0.46 on a random 5% subset of the data corresponding to the second row in Table 2 (data generated specifically for GPT-4o). On that same 5% of the data, models performed as follows:
> > >
> > >
> > > | Model                         | # Completion Tokens Budget | Accuracy | No Response Ratio
> > > | ----------------------------  | -------------------------- | -------- | -----------------
> > > | o1-preview                    | 1000                       | 46%      | 46%
> > > | Llama-3.1-70B-Instruct        | 40                         | 61%      | 0%
> > > | gpt-4o                        | 40                         | 58%      | 0%
> > > | Mixtral-8x7B-Instruct-v0.1    | 40                         | 40%      | 0%
> > >
> > > o1-preview's performance is worse than GPT-4o since it often does not output a response. o1-preview's accuracy might increase if given more budget, but we want to emphasize that the current budget is already 25x of what the other models were allocated.
> > >
> > > For the models' performance on 100% of the data, please refer to the second row of Table 2.
> > >
> > > We thank the reviewer for their consideration!

---

### Author Response · Authors · 2024-11-21
**General Response**

We thank the reviewers for their feedback! We are pleased to hear that reviewers found that our work tackles a “very important research question” [Reviewer XWk4]. Reviewers mention that our synthetic data generation framework creates a “robust training set” [Reviewer BPYe] with “strong generalization potential” [Reviewer BPYe; Reviewer rkK5] and thanks to its structure, data “correctness can be easily verified” [Reviewer p4tC]. TrackTheMind’s data can be also used as a “challenging benchmark to evaluate the mind reasoning capabilities of LLMs” [Reviewer BPYe; Reviewer XWk4] and it also “facilitates investigation into why mind reasoning tasks remain challenging” [BPYe] with some “unexpected and insightful” results [Reviewer rkK5] that may “indicate a mechanism different from that of humans in LLMs to tackle ToM tasks” [Reviewer rkK5].

**We address some common questions below, including several new analyses based on reviewers’ suggestions**: a detailed cost analysis of A* in terms of # tokens, o1-preview evaluation, and more detailed dataset statistics.

---
### _Additional statistics on TrackTheMind’s resulting dataset._
Stories shown in Table 1 were obtained by using one of 100 generated contexts, which were generated by first independently sampling a list of names and initial locations and then jointly sampling the full context. Table 1 stories contain 77 people in total, 28 objects, 47 containers, 52 rooms, 143 discussion topics. Object state updates ($a_{\text{updateObjState}}$) include 38 distinct visible and 26 invisible updates.

Story structures are diverse: if we consider the first 5 actions in each story (e.g. $[a_{\text{enterRoom}}, a_{\text{moveObjContainer}}, a_{\text{leaveRoom}}, a_{\text{enterRoom}}, a_{\text{moveObjRoom}}]$, without considering any variables), then a sequence of actions is repeated only 2.30 times on average (std=2.92; median is exactly 1). Story structures have 8.02 actions on average (std=2.37). Given that we request to infill each action with up to 2 sentences, resulting infilled stories are not long (avg token length=380, std=144), and there are 10K+ distinct tokens used.

---
### _o1-preview analysis, since inference-time scaling may boost performance_
Inference-time scaling will definitely help for ToM reasoning, as introduced in SymbolicToM (Sclar et al., 2023). As discussed in the intro, these high-cost inference-time solutions may not always be feasible, and o1 is an extreme example. Concretely, we evaluated o1-preview with 25x the completion tokens that we permitted to regular models on a sample of 500 TrackTheMind (story structure, question) pairs generated using GPT-4o. This is just 5% of the data analyzed in Table 2, and yet o1-preview evaluation costs ~$120. **We found that this extensive budget was not enough compute for o1-preview to produce _any final answer_ in 46% of the questions**. o1-preview was correct also for 46% of the questions; and plain incorrect for the remaining 8%.

---
### _On the cost of A* search_
Running A* with a budget of $N$ nodes (i.e., partial stories) costs exactly N times the cost of evaluating that node (story) if it were to be included as part of a final dataset. N can be modified as desired if efficiency is a concern. We selected $N=50$ since we believe that for the evaluation benchmark application it is vital to find the most challenging data points for evaluation and this outweighs efficiency concerns.

Since each node is a potentially shorter version of the final data point, and evaluating a final story over all first-order questions takes on average 681 input tokens (std=431) and 21 completion tokens (std=16), we can confidently say that A* with N = 50 nodes will take on average less than 681 * 50 = 34050 input tokens, and 21 * 50 = 1050 completion tokens. This relaxed upper bound totals $0.0478 per story generated adversarially for the frontier model GPT-4o.

---

[response continues in the following message]

---

> ### Author Response · Authors · 2024-11-21
> **General Response, Part 2**
>
> ### _Is complexity modulated by the # people and # actions? Why may a story with a greater number of people not necessarily imply a higher difficulty?_
> In TrackTheMind, the number of people and actions is important from a controllability and diversity perspective, but does not directly quantify task difficulty–difficulty quantification for ToM is an active area of research. Huang et al., 2024 quantifies a ToM problem complexity as the number of states necessary to solve it correctly (note that their approach requires manual annotation). While the number of states tends to increase with the number of people and actions, many questions do not require analyzing the whole story, e.g. if someone only entered the scene right at the end. When randomly sampling stories while fixing the number of core actions to e.g. 5, it’s more likely to have some characters with little involvement in the scene if there are 5 people in total than if there are 2 people. Since accuracy is computed across all questions about all characters, having a larger number of people may bump the average accuracy. TrackTheMind’s flexible framework allows for minimizing these cases through modifying the lookahead $h(s)$, but we chose against both doing this or filtering questions to show the performance is low even without these considerations.
>
> ---
>
> We look forward to discussing any remaining questions!
>
>
> References
> - Huang, XA, et al. A Notion of Complexity for Theory of Mind via Discrete World Models. EMNLP 2024.

---

### Meta-Review · Area_Chair_HB3a · 2024-12-18

**Metareview:**

This paper introduces a framework for generating synthetic theory of mind datasets using programmatic A* search over story structures. The framework produces diverse, challenging, and high-quality theory of mind evaluation and training data. The results highlight significant limitations of existing large language models (LLMs) in theory of mind reasoning and demonstrate that fine-tuning on TrackTheMind data improves performance on several benchmarks.

Reviewers generally agreed on the soundness of the experiments, particularly the analyses on the impact of interesting/uninteresting questions and the conceptual insights into LLM performance. The authors' response effectively clarified concerns, particularly around computational cost, dataset diversity, and complexity metrics, and provided additional experimental results.

Some reviewers raised concerns around new insights into ToM reasoning beyond confirming LLM limitations, bias of LLMs for generated context, and computational efficiency of the search. The authors provided clarifications and additional analyses (e.g., cost analysis, o1-preview evaluation), which addressed most concerns. All reviewers ultimately leaned towards acceptance.

**Additional Comments On Reviewer Discussion:**

During the discussion, reviewers raised several concerns, including the computational cost of A* search, potential biases in LLM-generated contexts, the simplicity of story structures for real-world applicability, and whether improvements on benchmarks reflect generalizable ToM reasoning. The authors addressed these by providing a detailed cost analysis, clarifying the design choices that mitigate bias, emphasizing the framework’s flexibility for richer scenarios, and explaining how TrackTheMind goes beyond prior work by enabling stress-testing and conceptual insights into ToM reasoning. They also added new experiments (e.g., o1-preview evaluation) and dataset statistics The responses effectively clarified key concerns from the reviewers.

---

### Decision · Program_Chairs · 2025-01-22

Accept (Poster)